# Epitope-directed monoclonal antibody production using a mixed antigen cocktail facilitates antibody characterization and validation

Oi Wah Liew [1,5✉], Samantha S. M. Ling[1,5], Shera Lilyanna[1], Yue Zhou[1], Peipei Wang[1], Jenny P. C. Chong[1], Yan Xia Ng[1], Angeline E. S. Lim[1], Eliot R. Y. Leong[1], Qifeng Lin [2], Teck Kwang Lim[2], Qingsong Lin[2], Enoch M. W. Ng[3], Tuck Wah Ng[3] & A. Mark Richards[1,4]

High quality, well-validated antibodies are needed to mitigate irreproducibility and clarify conflicting data in science. We describe an epitope-directed monoclonal antibody (mAb) production method that addresses issues of antibody quality, validation and utility. The workflow is illustrated by generating mAbs against multiple in silico-predicted epitopes on human ankyrin repeat domain 1 (hANKRD1) in a single hybridoma production cycle. Antigenic peptides (13–24 residues long) presented as three-copy inserts on the surface exposed loop of a thioredoxin carrier produced high affinity mAbs that are reactive to native and denatured hANKRD1. ELISA assay miniaturization afforded by novel DEXT microplates allowed rapid hybridoma screening with concomitant epitope identification. Antibodies against spatially distant sites on hANKRD1 facilitated validation schemes applicable to two-site ELISA, western blotting and immunocytochemistry. The use of short antigenic peptides of known sequence facilitated direct epitope mapping crucial for antibody characterization. This robust method motivates its ready adoption for other protein targets.

[1] Cardiovascular Research Institute, Department of Medicine, Yong Loo Lin School of Medicine, National University of Singapore, National University Health System, Singapore, Singapore. [2] Department of Biological Sciences, National University of Singapore, Singapore, Singapore. [3] Laboratory for Optics & Applied Mechanics, Department of Mechanical & Aerospace Engineering, Monash University, Clayton, VIC, Australia. [4] Christchurch Heart Institute, University of Otago, Christchurch, New Zealand. [5] These authors contributed equally: Oi Wah Liew, Samantha S. M. Ling. ✉email: mdclow@nus.edu.sg

Antibodies are protein workhorses with broad utility in research, diagnostic and therapeutic applications. However, performance inconsistencies and poor validation are often encountered with commercial antibodies, contributing to irreproducible and misleading data[1–6]. For example, inadequate antibody characterization have led to controversies surrounding the role of growth differentiation factor 11 (GDF11) in age-related cardiac, muscle and cognitive decline[7]. The antibody used in early high-profile reports was later found to cross-react with a closely-related family member, myostatin (GDF8), raising concerns over the validity of the original findings[8,9]. High quality antibodies are also urgently needed for large-scale initiatives like the Human Atlas Project to map organ, tissue and cellular distribution of proteins in the human proteome. Corroboration with at least two validated antibodies to non-overlapping epitopes on the same protein is required for data reliability[10]. To meet the expanding need for fit-for-purpose antibodies, we developed a peptide-mediated method for producing high affinity monoclonal antibodies (mAbs) against multiple sites on a target protein. The robust workflow lowers the barrier to quality mAb production by dramatically reducing the experimental burden associated with hybridoma screening, antibody characterization and validation.

Whole proteins have traditionally been used as immunogens for antibody production. Since the seminal work of Arnon and colleagues showing that protein-reactive antibodies could be elicited by a synthetic peptide[11], the versatility of peptide polyclonal and monoclonal antibodies has led to increasing interest in using protein fragments (100–150 residues) and peptides (10–20 residues) as surrogates of whole proteins[12–14]. Studies have shown that peptides can produce antibodies directed towards protein domains[13,15], single amino acid mutations[16] or highly specific post-translational modifications[17–19] that were not possible or feasible with whole proteins as immunogens. In another documented case, peptides were more efficient compared with the intact protein in producing antibodies with specificities towards both peptide and intact protein[20]. These advantages along with the relative ease for epitope mapping have provided strong impetus for using peptides as immunogens for antibody production. To ensure immunogenicity, synthetic peptides may be chemically conjugated to large carrier proteins such as bovine serum albumin (BSA), ovalbumin or keyhole limpet hemocyanin (KLH), or prepared as branched peptides known as multiple antigen peptides (MAP). Over the past few decades, sustained development of peptide antibodies have been facilitated by rapid growth in protein sequence and structural information along with concurrent advances in peptide synthesis and conjugation, bioinformatics tools for epitope prediction and peptide selection, and epitope mapping strategies for antibody characterization[14].

We advance an epitope-directed approach to generate anti-peptide mAbs targeting several non-overlapping protein sites in a single hybridoma production cycles. Short, spatially distant, B-cell epitope-predicted sequences are independently cloned into the surface exposed loop of a highly soluble His-tagged thioredoxin (Trx) carrier. This facilitates high-yield production and easy purification of bacterially expressed fusion peptides, which are then combined into a mixed immunogen cocktail for animal immunization. DEXT microplates (based on previously reported 'transparency' or flat sheet designs[21–23]) manufactured to the dimensional footprint of a 96-well microplate but requiring only 15 μL per well (instead of typical 50–100 μl volumes in conventional microplate assays) allowed rapid screening of hybridoma clones with concomitant epitope identification by ELISA. Unlike with whole-protein immunogens where downstream epitope mapping involves scanning of the entire target protein sequence, the use of peptide antigens confines epitope analyses to short known sequences, greatly reducing the size of the peptide library

required for identifying the epitope footprint and critical binding residues. Generating antibody panels to multiple non-overlapping sites supports validation schemes based on 'independent antibody assessment'. This also facilitates two-site ELISA development, an assay format that relies on non-overlapping dual antibody recognition of the target protein and considered to enhance assay sensitivity and specificity as well as compatibility with complex sample matrices compared to single antibody formats.

Human ankyrin repeat domain 1 (hANKRD1) was selected as the test protein to demonstrate the utility of our method. This molecule is a component of the cardiac sarcomere, exhibits cytosolic-nuclear shuttling and plays pivotal roles in transcriptional regulation, sarcomere assembly and mechano-sensing in the heart (reviewed in ref. [24]). hANKRD1 is almost exclusively expressed in cardiac muscle and interacts with a plethora of functional and structural proteins[25,26]. It is associated with regulatory networks that govern hypertrophic, fibrotic and apoptotic responses to cardiac stress. As with many molecules of interest, the lack of validated antibodies against hANKRD1 has contributed to the dearth of information on its expression profile, functions, regulation and clinical significance at the protein level. The availability of well-characterized mAbs directed towards functionally relevant protein domains would facilitate mechanistic studies of hANKRD1 processing, cellular localization, binding profile with interacting partners and pathophysiological dysregulation within the cell and in circulation.

Using the described method, we produced a panel of mAbs of picomolar affinity that target the N-, middle and C-terminal regions of hANKRD1. Epitope information on the critical binding residues predicted reactivity of some mAbs to rat ANKRD1 (rANKRD1). Utility of the mAbs were experimentally confirmed for ELISA, western blotting, immunoprecipitation and immunocytochemistry applications. This method will be of general applicability for producing 'protein-reactive' anti-peptide mAbs of predetermined specificities to protein targets; their wide ranging applications in biology and medicine providing a powerful rationale for adopting this approach for antibody panel generation[27].

## Results

**Principle of method.** Figure 1 provides a schematic overview of the method and associated advantages. Antigenic sites on hANKRD1 were first identified using the Bepipred B-cell linear epitope prediction tool[28–31] (Fig. 2a, b). The positions of the selected antigenic sequences (designated AG1, AG4 and AG5) are shown with respect to the predicted 3-D model and linear amino acid sequence (Fig. 2c, d). AG2 and AG3, which overlaps with or are in close proximity to protein interaction sites, were not selected as antibody binding may potentially be hindered. AG1, −4 and −5 were individually displayed as tandem three-copy inserts on the surface exposed loop of the His-tagged Trx scaffold (Fig. 2e). All Trx-tripeptide proteins were produced in *Escherichia coli* and highly expressed in the soluble form accounting for 20–30% of total bacterial protein (Supplementary Note 1 and Supplementary Fig. 1). The antigens were purified to >95% purity by 1-step native immobilized metal affinity chromatography (IMAC). Protein yields ranged from 37 to 41 mg/L bacterial culture (Supplementary Table 1). These three antigens were then mixed in equimolar concentrations and used as an immunogen cocktail for animal immunization.

**Hybridoma screening and selection.** A single-fusion experiment of splenocytes from two mouse spleens with SP2/0-Ag14 myeloma cells produced a total of 2976 hybridoma clones. Initial screening was performed using validated 96-well DEXT

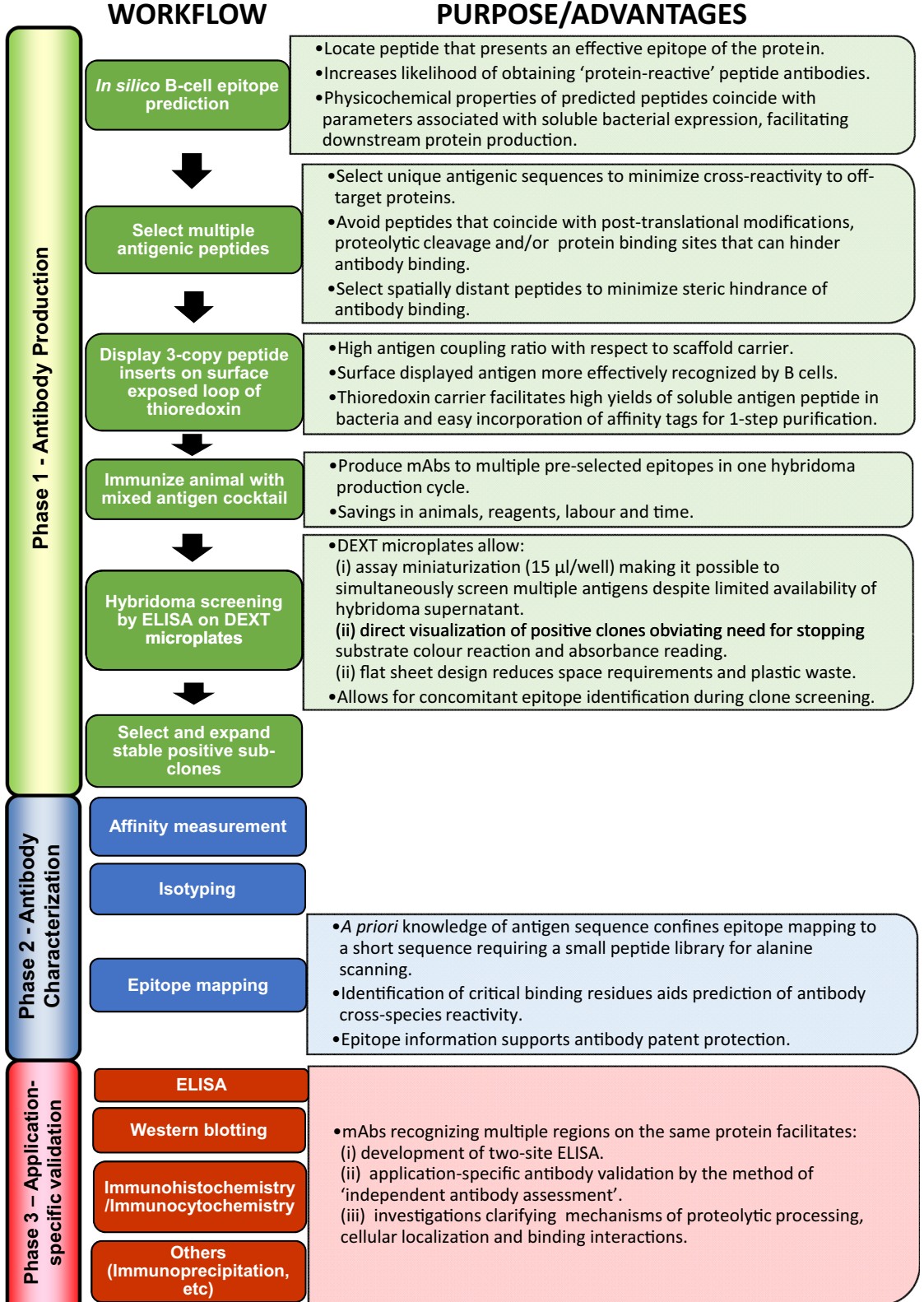

**Fig. 1 Three-phase workflow for generating, characterizing and validating mAbs.** Corresponding advantages/purpose at each step are highlighted.

microplates (Fig. 3a). Mean whole-plate coefficient of variation (CV) did not exceed 6% with antigen coating concentration at 1 μg/ml or above and this assay performance was comparable with conventional microplates (Fig. 3b). Evaporative losses were also kept at a minimum despite the vastly reduced assay volume (Supplementary Note 2 and Supplementary Fig. 2). Hybridoma clones producing mAbs that reacted strongly to only one of the three coated antigens were selected for further expansion

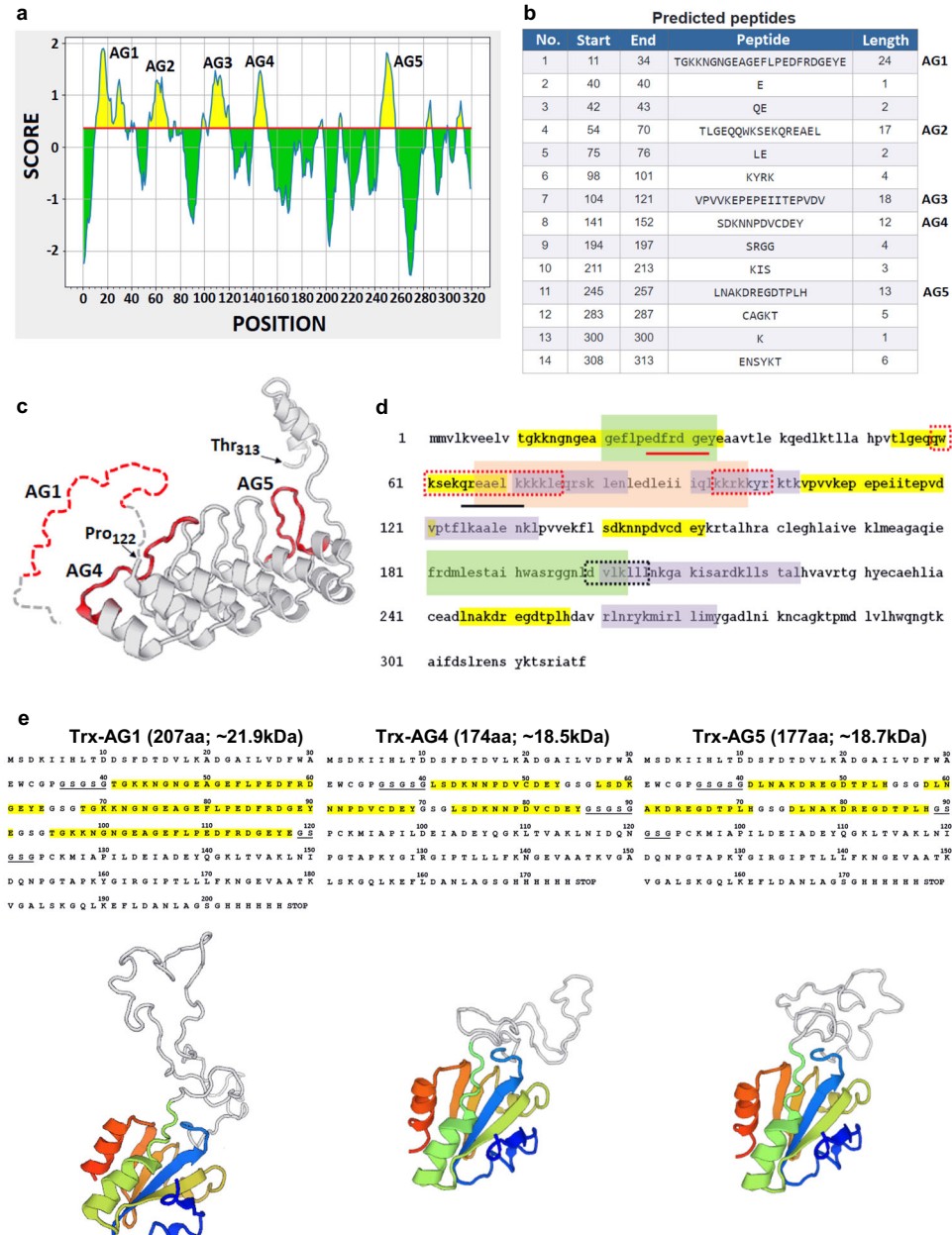

**Fig. 2 Peptide selection and antigen design. a** Output graph of predicted B-cell linear epitopes showing amino acid position (*x*-axis) and Bepipred score (*Y*-axis). Residues (yellow) with scores above the default threshold (red line) of 0.35 have a higher probability to be part of an epitope. **b** Five predicted epitopes (AG1–AG5) of lengths ≥12 amino acids (start and end positions indicated). **c** Swiss model of hANKRD1 from Pro$_{122}$ to Thr$_{313}$ using ANK-N5C-281 (PDB entry 4qfv.2.B) as template. Antigen sequences, AG4 and AG5, are shown in red. The sequence upstream of Pro$_{122}$ could not be modeled. A schematic representation of this region harboring AG1 is depicted by dotted lines. **d** Overlay of predicted antigenic sequences (yellow) against titin (green) and calsequestrin (purple) binding sites and the coiled coil region (orange) on hANKRD1. Putative nuclear localization (red dotted boxes), nuclear exporting signals (black dotted box), caspase 3 (red line) and calpain 3 (black line) proteolytic cleavage sites are shown. **e** Amino acid sequences of Trx-fused immunogens harboring three-copy inserts of antigen sequences (yellow) flanked by long GSGSG linkers (black underlined). The middle insert is separated from its flanking counterparts by shorter GSG linkers. The corresponding 3D structure predicted by homology modeling against thioredoxin 1 (1oaz.1.A.) as template are shown below each sequence. The coloured regions correspond to thioredoxin from its amino terminus (deep blue) to its carboxy terminus (red). The regions in white comprise the three-copy peptide inserts and flanking GSGSG linkers.

(Fig. 3c). A total of 74, 23 and 61 clones were found to react strongly to AG1, −4 or −5, respectively. Of these, ~80% were unstable and exhibited significant loss of reactivity after a second round of sub-culturing. Finally, a total of 24 stable primary parent clones of high specificity to their cognate antigen were obtained (Fig. 3d). Twelve stable high-yielding hybridoma cell lines were subcloned for further analysis. Antibodies were purified by

Protein A affinity chromatography with yields ranging between 23 and125 μg/ml of hybridoma culture supernatant.

**Monoclonal antibody characterization.** Table 1 summarizes the functional characteristics of the selected mAbs. Isotype analysis revealed a predominance of IgG-kappa antibodies and one

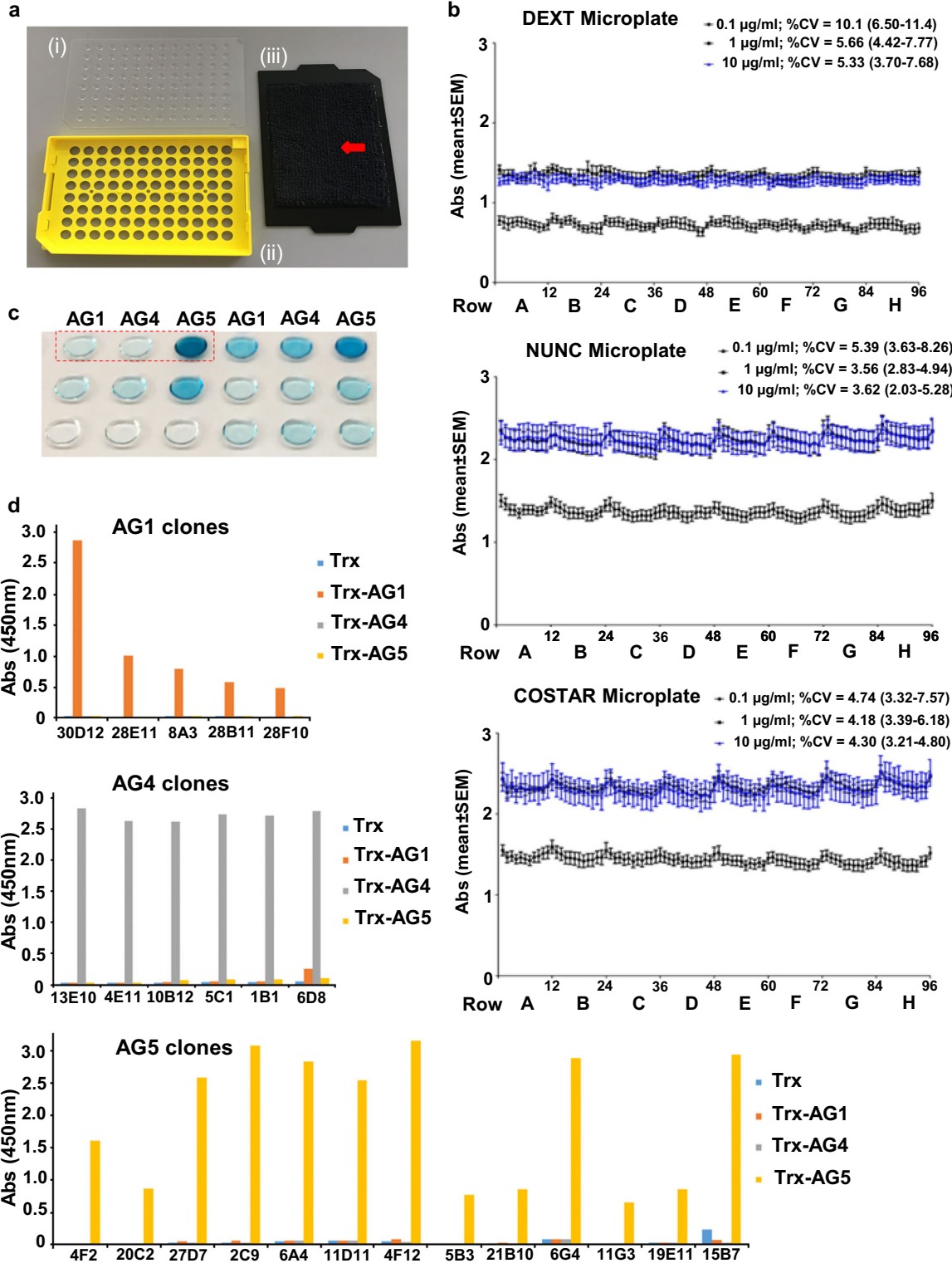

IgM-kappa. Binding of the antibodies to recombinant hANKRD1 (OriGene Technologies) and their cognate Trx-fused peptide were evaluated by surface plasmon resonance (SPR) analysis (Supplementary Note 3 and Supplementary Fig. 3). Ten mAbs were able to interact with both the protein and peptide. AG4/mAb 13E10 and 4E1 failed to bind to hANKRD1 but were able to bind to Trx-AG4. The monovalent binding affinities (KD) of the mAbs to hANKRD1 were at least 0.4 nM or better. Epitope mapping by alanine scanning allowed direct identification of

residues that are critical for antigen–antibody binding [Table 1; Supplementary Fig. 4]. Since short antigen sequences are used, only a small library of 54 peptides (including the three wild-type peptides as control) were needed. The functional epitopes of the mAbs comprise 1, 3 or 4 key interacting residues.

Amino acid sequence alignment of hANKRD1 and rANKRD1 shows full conservation for AG5, one residue mismatch for AG4 and multiple residue mismatches for AG1 (Supplementary Fig. 5). It is predicted that hANKRD1-reactive

**Fig. 3 Hybridoma screening using DEXT microplates. a** Assemblage of DEXT microplates showing (i) flat APET sheets with scribed wells, (ii) re-useable custom-made base, and (iii) re-useable lid with affixed microfiber cloth (red arrow). **b** Scatter plot of mean absorbance (vertical bars showing standard error of mean) against well position at three antigen coating concentrations obtained from DEXT, NUNC and COSTAR microplates. Data are presented in sequential order by rows (indicated by alphabets A–H) followed by columns. Abs arbitrary optical density units at 450 nm (background subtracted), % CV mean (range) whole-plate coefficient of variation ($n = 7$ plates per concentration tested). **c** A section of the DEXT microplate to illustrate the selection process for six hybridoma clones and concomitant epitope identification. Columns of wells are coated with the antigen (1 μg/ml) indicated at the top of the panel. Culture supernatant (15 μl per well) from a hybridoma clone was added to three wells coated individually with Trx-AG1, -AG4 or -AG5. The positive clone selected for further expansion, antibody purification and characterization is indicated by the red dash box. This clone demonstrates very strong (intense blue colour) and highly specific reaction to one antigen and the epitope it binds to is immediately identified as AG5. **d** Screening of stable parent clones that are specific to AG1, AG4 and AG5 by ELISA. Each mAb was tested against thioredoxin (Trx), Trx-AG1, Trx-AG4 and Trx-AG5. All clones gave absorbance values from their cognate targets that were 20–100 times above the mean absorbance of the off-target background.

**Table 1 mAb characterization by isotyping, SPR and alanine scan analysis.**

| Target | Hybridoma | Isotype | $k_a$ (M$^{-1}$s$^{-1}$) | $k_d$ (s$^{-1}$) | KD (M) | Critical residues |
|---|---|---|---|---|---|---|
| AG1 | 30D12 | IgG$_1$/kappa | $4.51 \times 10^5$ | $2.05 \times 10^{-5}$ | $4.55 \times 10^{-11}$ | TGKKNGNGEAGEFLPED**FR**DG**EY**E |
| | 8A3 | IgM/kappa | $2.08 \times 10^5$ | $3.69 \times 10^{-5}$ | $1.77 \times 10^{-10}$ | TGKKNGNG**E**A**GE**FLPED**F**RDGEYE |
| AG4 | 13E10 | IgG$_1$/kappa | NB | NB | NB | NB |
| | 4E1 | IgG$_1$/kappa | NB | NB | NB | NB |
| | 10B12 | IgG$_{2A}$/kappa | $7.28 \times 10^5$ | $1.26 \times 10^{-4}$ | $1.73 \times 10^{-10}$ | LSD**KNNP**DVCDEY |
| | 5C1 | IgG$_1$/kappa | $3.07 \times 10^5$ | $3.56 \times 10^{-5}$ | $1.16 \times 10^{-10}$ | LSD**K**NNPDVCDEY |
| | 1B1 | IgG$_1$/kappa | $3.09 \times 10^5$ | $3.13 \times 10^{-5}$ | $1.01 \times 10^{-10}$ | LSD**K**NNPDVCDEY |
| AG5 | 4F2 | IgG$_1$/kappa | $6.68 \times 10^5$ | $1.84 \times 10^{-5}$ | $2.76 \times 10^{-11}$ | DLNA**KD**R**EG**D**T**PLH |
| | 27D7 | IgG$_1$/kappa | $7.05 \times 10^5$ | $1.23 \times 10^{-4}$ | $1.74 \times 10^{-10}$ | DLNAKDREG**D**T**PLH** |
| | 2C9 | IgG$_{2A}$/kappa | $4.47 \times 10^5$ | $8.14 \times 10^{-5}$ | $1.82 \times 10^{-10}$ | DLNAKDREG**DT**PL**H** |
| | 6A4 | IgG$_1$/kappa | $8.72 \times 10^5$ | $3.33 \times 10^{-5}$ | $3.82 \times 10^{-10}$ | DLNAKD**R**EGDTPLH |
| | 11D11 | IgG$_1$/kappa | $1.00 \times 10^6$ | $1.33 \times 10^{-5}$ | $1.33 \times 10^{-11}$ | DLNAKD**R**EGDTPLH |

Critical residues (highlighted in bold underline) are defined as those that reduce the ELISA colorimetric response to <50% of the wild-type sequence (Supplementary Fig. 4).
ka association constant, kd dissociation constant, KD equilibrium constant, NB no binding.

mAbs could bind to rANKRD1 except for AG1/mAb 8A3 and AG4/mAb 10B12 where amino acid mismatches coincide with the critical residues for those epitopes.

**ELISA**. Fifteen antibody pairs were found from $12 \times 12$ capture/detector mAb combinations by ELISA using recombinant hANKRD1 as test analyte (Supplementary Table 2). The antibody pair that gave the highest signal response comprised AG1/mAb 30D12 as capture and AG5/mAb 4F2 as detector while the reverse pairing resulted in the best signal-to-noise ratio. Only AG1/mAb 30D12 and AG5/mAb 4F2 displayed dual utility as both capture and detection antibodies while other protein-reactive mAbs are useful only as capture antibodies. Self-pairing was also observed for AG1/mAb 30D12 and AG5/mAb 4F2, suggesting the presence of dimeric hANKRD1 in the bacterially derived recombinant preparation. This was confirmed by SDS-PAGE analysis where both the 43-kDa monomeric and 86-kDa dimeric bands were observed under non-reducing conditions (Supplementary Fig. 6). None of the mAbs to AG4 were able to self-pair, presumably due to steric hindrance of antibody binding when hANKRD1 homo-dimerizes at its coiled coil region. All positive antibody pairs demonstrated a linear dose-response relationship over a concentration range of 0.31–20 ng/ml with R-squared values exceeding 0.99 (Fig. 4a).

To further validate mAb specificity, full-length hANKRD1 was expressed in *E. coli* from its synthetic cDNA cloned into pJ404. Directional insertion of this cDNA into 5′ Not I and 3′ Sal I sites of the eukaryote expression vector, pCMV-Myc, positioned the hANKRD1 sequence downstream of the myc tag (designated pCMV-Myc/hANKRD1). ELISA data obtained with the 4F2/30D12 antibody pair showed parallel dilutional linearity (recovery between 90 and 123% within the standard curve range) in soluble lysates (starting total protein concentration at ~2 mg/ml)

prepared from induced *E. coli* and transfected H9c2 cells (Fig. 4b). This is strong indication of comparable selectivity between endogenous rANKRD1 and vector-derived hANKRD1. The concentration of endogenous rANKRD1 in H9c2 cells transfected with pCMV-Myc was 4.54 ng/ml. As expected, transfection with pCMV-Myc/hANKRD1 resulted in increased ANKRD1 (rANRKD1 plus ectopic hANKRD1) concentration (10.3 ng/ml). Blocking of the detection antibody with the AG1 matched peptide completely abolished hANKRD1 signal in *E. coli* lysate while signal retention was observed when blocked with the AG4 unmatched peptide (Fig. 4c). Similar trends were observed with transfected H9c2 lysates although some residual signal was still present after blocking with the matched AG1 peptide.

We show that the 4F2/30D12 antibody pair can be applied in ELISA to measure plasma rANKRD1. The assay has a limit of detection (defined as the concentration derived from the mean optical density of 20 zero standard replicates plus 3 standard deviations) of 0.073 ng/ml. Myocardial infarction (MI) was induced in an experimental group of rats by ligation of the left anterior descending coronary artery while the same surgical procedure without ligation was performed on the Sham group. EDTA plasma samples collected on day 7 and 30 were assayed. Day-7 median (interquartile range) rANKRD1 concentration in MI was 0.60 ng/ml (0.52–0.68 ng/ml) and was significantly lower ($p = 0.0495$) compared with Sham [0.71 (0.62–0.83) ng/ml] (Fig. 4d). An apparent restoration of plasma rANKRD1 was observed by Day 30 in MI rats where levels were no longer significantly different from Sham.

**Western blot**. Western blotting showed antibody reactivity towards recombinant hANKRD1 and endogenous rANKRD1 in lysates of rat cardiac tissue and neonatal cardiomyocytes (Fig. 5a; Supplementary Fig. 7a,b). Nine mAbs detected hANKRD1 at the

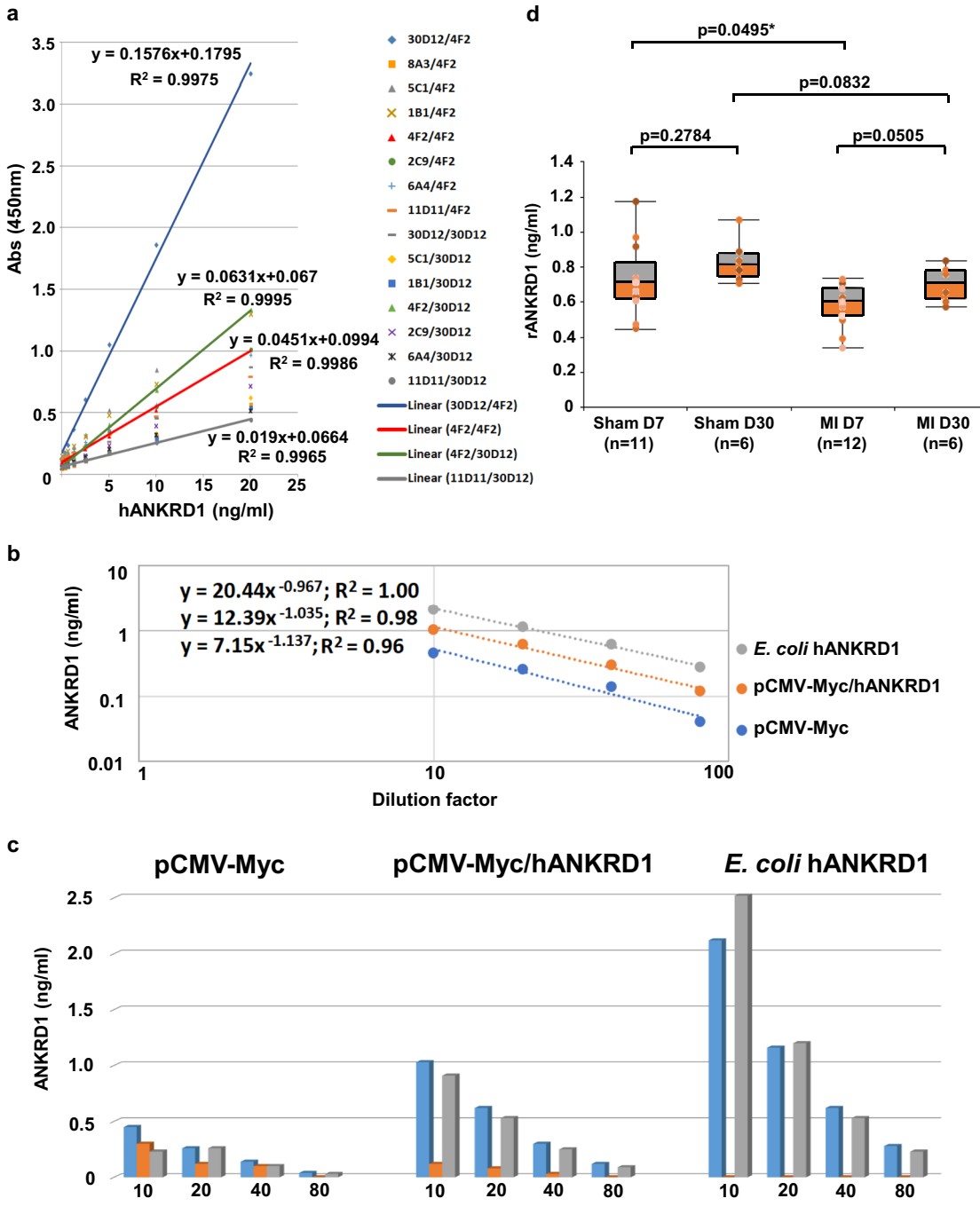

**Fig. 4 Antibody validation by ELISA. a** Antibody-pair dose response against recombinant hANKRD1 (OriGene Technologies). The legend lists each capture/detection antibody matched pair used. A linear curve fit over analyte concentration ranging from 0.313–20 ng/ml is applied to representative antibody pairs. **b** Dilution curves (fitted with power-law regression where the exponent of function x represents the slope of the curve) of endogenous rANKRD1 and ectopic Myc-hANKRD1 present in transfected H9c2 lysate and hANKRD1 from induced *E. coli* cells. Data points represent 10, 20, 40 and 80 fold dilution of cell lysates with starting total protein concentration at ~2 mg/ml. Both axes are plotted in the log10 scale for better visualization of parallelism. **c** Effect of peptide competition in ELISA where the detection antibody is blocked by matched or unmatched peptides. Data from 10, 20, 40 and 80 fold diluted lysates are shown. **d** Plasma rANKRD1 concentration (ng/ml) in experimental MI and Sham rat on days 7 (D7) and 30 (D30). The boxplot represents data quartiles (25th, median and 75th percentile) and error bars represent the maximum and minimum data values. Data distribution for each boxplot is indicated by the coloured dots. Two-tailed unpaired *t*-test was used to compare each two-group combination with significance at *p* < 0.05. n number of individual animals in each group.

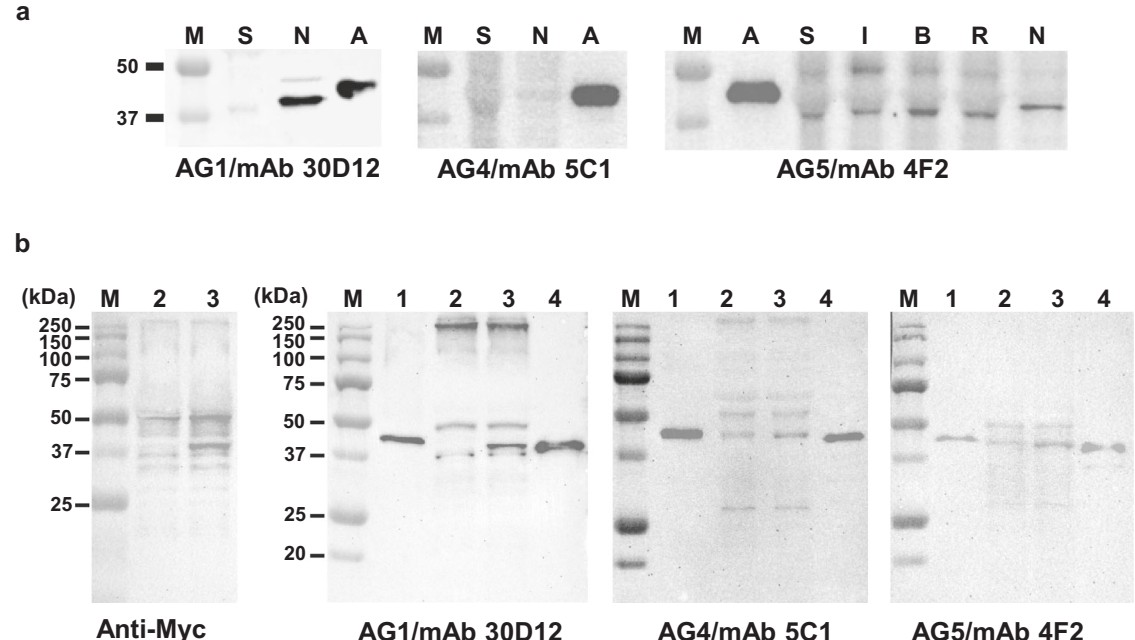

**Fig. 5 Antibody validation by western blotting. a** Recombinant hANKRD1 (A; 10 ng per well) and total protein (50 μg per well) isolated from whole-cell lysates of adult rat heart tissue (Sham, S), myocardial infarcted Day 2 tissue (infarcted region, I; border region, B; remote region, R) and neonatal rat cardiomyocytes (N), were separated by SDS-PAGE, transblotted onto PVDF membranes and probed with antibody indicated below each panel. M, All-Blue Precision Protein Standards (BioRad) showing only the 50 and 37 kDa marker bands. **b** Western blot analysis of lane 1. purified recombinant hANKRD1 (Origene Technologies, 4 ng/well); lane 2. lysate from pCMV-Myc transfected H9c2 cells; lane 3. lysate from pCMV-Myc/hANKRD1-transfected H9c2 cells; lane 4. lysate from *E. coli* cells expressing hANKRD1, probed with antibodies indicated at the bottom of each panel. Total protein loading for H9c2 and *E. coli* lysates were 20 μg per well.

43.9 ± 0.8 kDa band position and this was concordant with the expected mass of 43 kDa as indicated in the manufacturer's technical data sheet. Of these, 5 mAbs also detected rANKRD1 at molecular weight positions of 39.0 ± 0.7 and 41.0 ± 0.3 kDa in adult cardiac tissue and neonatal cardiomyocytes, respectively. This is in close agreement with the previously reported 40-kDa molecular mass of ANKRD1 isolated from rat heart muscle[32] and rat neonatal cardiomyocytes[33,34].

The ability of the mAbs to recognize rANKRD1 is generally concordant with the epitope mapping results. Critical residues for AG1/mAb 30D12 binding are conserved between the human and rat sequence, thus allowing antibody binding in both species. Similarly, AG5 mAbs (except for 27D7, which did not detect hANKRD1 by western blotting although strong binding was observed by SPR) were able to bind rANKRD1 since the AG5 sequence exactly matches that of rANKRD1. As expected, AG1/mAb 8A3 and AG4/mAb 10B12 did not recognize rANKRD1 as critical residues for epitope binding coincided with the amino acid mismatch between the human and rat sequence. However, hANKRD1-reactive AG4/mAb 5C1 and 1B1 were not able to detect rANKRD1 although there was no sequence mismatch at the positions of their critical residues. Three mAbs (AG4/mAb 13E10, AG4/mAb 4E1 and AG5/mAb 27D7) gave negative results for hANKRD1 on Western blots (Supplementary Fig. 7c). However, these mAbs detected single bands of unknown origin at the 43.0 ± 1.05 and 44.1 ± 0.4 kDa positions in rat adult cardiac tissue and neonatal cardiomyocytes, respectively. For comparison, a western blot was also performed using a commercial polyclonal antibody (Atlas Antibodies, HPA038736) raised against the C-terminal 52-residue fragment of hANKRD1. This antibody is validated only for immunohistochemistry /immunocytochemistry applications but not western blotting where no immunodetected bands were obtained from a panel of

human tissue and cell line samples tested within the Human Protein Atlas project. In our study, HPA038736 was found to be non-reactive to hANKRD1 and endogenous rANKRD1 in rat neonatal cardiomyocytes by western blot analysis (Supplementary Fig. 8). However, a similar banding pattern to that obtained for AG4/mAb 4F2 in adult rat cardiac tissue was observed with this antibody. We also demonstrated by western blotting and antigen-down assay that AG1/mAb 30D12, AG4/mAb 5C1 and AG5/mAb 4F2 do not cross-react with hANKRD2, a closely-related MARP family member that shares high sequence and functional similarities with hANKRD1 (Supplementary Figs. 9 and 10).

In transfection experiments, ectopic hANKRD1 at the expected ~40-kDa position was observed in pCMV-Myc /hANKRD1 but not pCMV-Myc transfected H9c2 lysate when probed with an anti-Myc antibody and AG1/mAb 30D12 (Fig. 5b). Surprisingly, with AG4/mAb 5C1 and AG5/mAb 4F2, an immunodetected band that coincides with the 40-kDa position of full-length endogenous rANKRD1 was detected in pCMV-Myc transfected lysate while intensity of the equivalent band in pCMV-Myc /hANKRD1 lysate was slightly higher, indicative of ectopic hANKRD1 expression. All three mAbs also detected a 50-kDa band, the identity of which remains to be clarified.

**Immunoprecipitation.** Immunoprecipitation from soluble lysates derived from hANKRD1-expressing *E. coli* (14.1 mg total protein), neonatal rat cardiomyocytes (1.13 mg total protein) and pCMV-Myc/hANKRD1-transfected H9c2 cells (1.32 mg total protein) was performed using biotinylated AG5/mAb 4F2/strep-tavidin magnetic beads and analysed by mass spectrometry after tryptic digestion of the antibody pull-down eluates. hANKRD1 was identified with six peptides among the 283 proteins (confidence level >95%) detected in *E. coli* lysate (Supplementary Fig. 5). Neither rANKRD1 nor hANKRD1 were detected in the

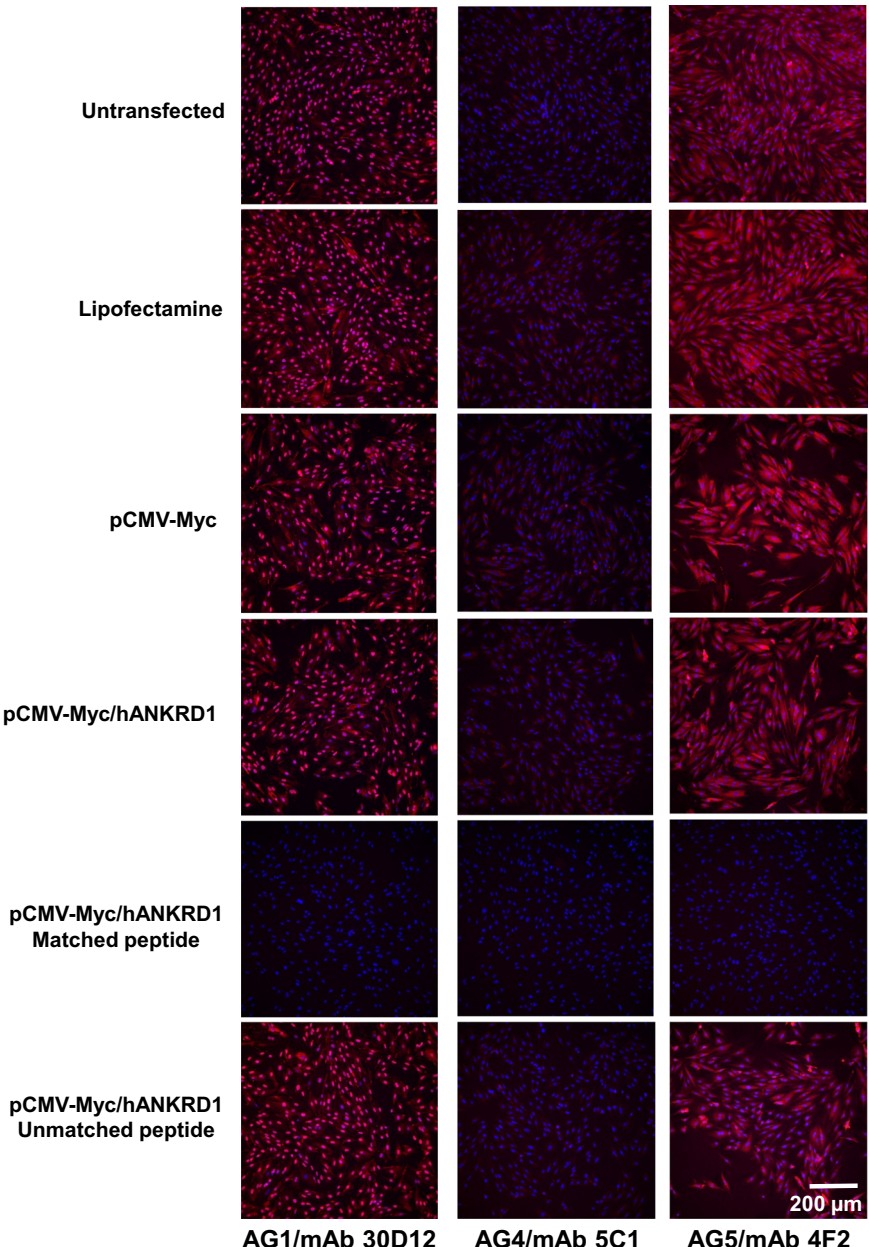

**Fig. 6 Immunofluorescence staining of H9c2 cells under different treatment conditions.** Panels represent merged images of cells stained for ANKRD1 (red) and nucleus (DAPI, blue). The antibodies used are indicated below the bottom panels. Blocking with the corresponding matched peptide of each mAb completely abolished staining for ANKRD1 in pCMV-Myc/hANKRD1-transfected cells. In contrast, blocking with the unmatched peptide for AG1/mAb 30D12 (AG4 peptide), AG4/mAb 5C1 (AG1 peptide) and AG5/mAb/ 4F2 (AG1 peptide) resulted in marginal loss of ANKRD1 signal.

rat IP eluates, likely due to a combination of lower concentration of ANKRD1 present compared with *E. coli* lysate, lower input rat lysate applied and sensitivity of the MS instrument used.

**Immunostaining**. AG1/mAb 30D12, AG4/mAb 5C1 and AG5/ mAb 4F2 were tested for their utility in immunofluorescence cytochemistry (IFC). Untransfected and transfected H9c2 cells showed primarily nuclear localization of ANKRD1 with AG1/ mAb 30D12, perinuclear localization with AG4/mAb 5C1 and cytoplasmic distribution with AG5/mAb 4F2. Blocking of the antibody probes with corresponding matched peptides abolished ANKRD1 staining but not with unmatched peptides (Fig. 6).

Transverse sections from the mid-section of post-MI day-7 rat heart were immunostained and the staining pattern of cells

oriented longitudinally and transversely are compared (Fig. 7). Immunostaining by AG1/mAb 30D12 and AG5/mAb 4F2 revealed more intense fluorescence signals in the border zone of MI tissue compared to the remote regions, suggesting an upregulation of ANKRD1 expression in border cells. A differential staining pattern between cells oriented longitudinally and transversely was observed where the striations of the myofibrils are more distinct in the former orientation. Myofibrillar striations were also more evident by immunostaining with AG1/mAb 30D12 than with AG5/mAb 4F2. While AG1/mAb 30D12 detected rANKRD1 in both the nuclear and cytoplasmic regions of myocytes, AG5/mAb 4F2 detected rANKRD1 only in the cytoplasm. Negative staining was observed for AG4/mAb 5C1 and this is in agreement with western blot data from rat cardiac tissue.

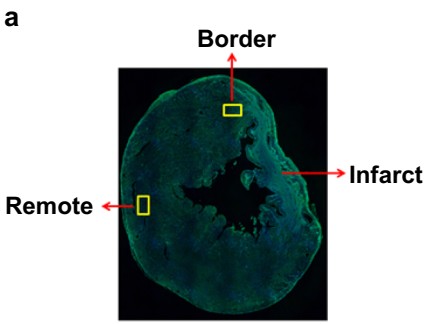

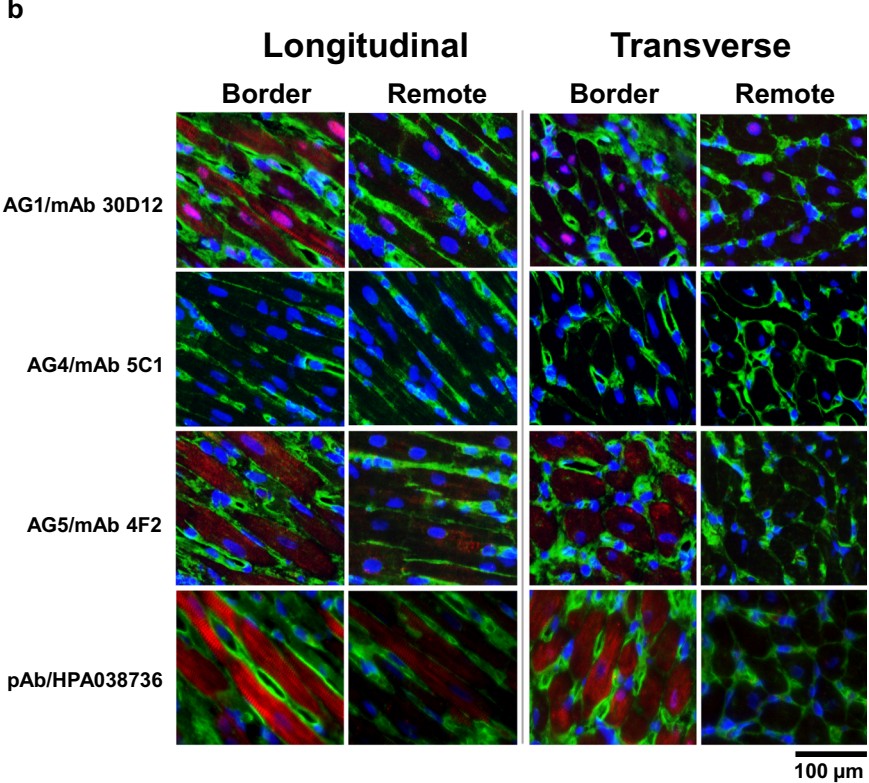

**Fig. 7 Rat ANKRD1 is localized in the nucleus and sarcomere, and is upregulated in response to cardiac insult. a** Transverse section of rat heart tissue (post-MI day 7) showing the border, remote and infarcted zones. **b** Immunofluorescence staining for rANKRD1 in post-MI day-7 rat heart tissue. The regions that are at the border or remote from the infarcted zone are examined in the longitudinal and transverse orientation. rANKRD1 is labeled by Alexa Fluor 594 (red), the cytoplasm by Alexa Fluor 488 (green) and nucleus by DAPI (blue). The merged image reveals that rANKRD1 is upregulated in the border region and is present in both the cytoplasm (localized in the myofibrils) and nucleus of myocytes.

Immunostaining with HPA038736 recapitulates the pattern observed with AG5/mAb 4F2, although the fluorescence signal from the former was of much stronger intensity.

## Discussion

A systematic workflow for producing high affinity mAbs directed against pre-selected peptides (13–24 residues) of a protein of interest is described. A cocktail of three Trx-tripeptide antigens corresponding to short sequences in the N-, C- and internal regions of hANKRD1 yielded 24 stable hybridoma clones and 12 mAbs were selected for detailed evaluation and validation. Ten mAbs demonstrated protein-reactivity towards hANKRD1 by ELISA and SPR. Fifteen antibody pairs suitable for two-site ELISA were obtained. Nine mAbs detected hANKRD1 by western blotting. This equates to generating one hANKRD1-reactive mAb suitable for ELISA and western blot applications for every 331

initial clones screened, although the success rate may be higher with rigorous testing of all 24 stable clones. This compares well with the estimation of one mAb suitable for immunoprecipitation of the target protein for every 242 initial clones screened as reported in a large consortium effort generating mAbs using either hybridoma supported by automated high-throughput screening technologies or phage display approaches[35]. We extracted the available curated binding affinity data of 87 mouse monoclonal antibodies from the SabDab structural antibody database[36] and used the published KD values to classify their binding strength as low (micromolar range; $10^{-5}–10^{-6}$ M), moderate (nanomolar range; $10^{-7}–10^{-9}$ M), high (picomolar range; $10^{-10}–10^{-11}$ M) and very high ($10^{-12}$ M). The minimum affinity requirement of functional (or practically useful) antibodies lies in the nanomolar range and we found that 82% of antibodies in the database were of low to moderate binding affinity. In contrast, 10 out of 12 peptide mAbs (or 83%)

produced in this work displayed picomolar affinity for the native protein. This efficiency is higher or comparable to previous reports where between 25% (1 out of 4) to 100% (2 out of 2) of mAbs[37] and 56% of polyclonal serum[38] generated from KLH-conjugated peptides were protein-reactive. Identification of critical binding residues facilitated prediction of antibody cross-species reactivity. Evidence from ELISA, western blot and immunostaining data obtained from control and hANKRD1-transfected H9c2 cells as well as peptide competition experiments provide strong evidence for mAb specificity towards ANKRD1. AG5/mAb 4F2 mediated pull-down from hANKRD1-expressing *E. coli* cell lysate identified 6 tryptic-digested hANKRD1 peptides.

Each step of the workflow provides distinct advantages over traditional methods using whole proteins or carrier-coupled synthetic peptides as immunogens. Rational peptide selection begins with in silico epitope prediction. B-cell linear epitope prediction algorithms are trained to locate surface exposed sites on the target protein, increasing the likelihood of generating antibodies with utility over a broader range of applications. Additionally, predicted epitopes tend to be hydrophilic, solvent accessible and flexible[39]. These physicochemical properties coincide with parameters associated with protein solubility for heterologous bacterial expression, facilitating downstream recombinant peptide production[40]. Depending on the experimental objectives, factors for final peptide selection may include: (1) remoteness from modification sites to enhance antibody tolerance to post-translational changes in the native protein; (2) non-overlap with proteolytic cleavage and/or protein interaction sites to minimize interference with antibody binding; (3) surface display on the target protein to increase likelihood of antibody accessibility; (4) uniqueness of the peptide sequence to minimize cross-reactivity to off-target proteins. These considerations are illustrated with hANKRD1 whereby the predicted antigenic regions were superimposed over known protein interaction and protease cleavage sites. Peptides coinciding with or in close proximity to known titin/calsequestrin binding sites, the coiled coil region of hANKRD1 (the site for self-interaction to form homodimers)[41] and calpain/caspase cleavage sites are avoided. Since protein-reactive antigenicity is a key requirement, the locations of the selected peptides on the hANKRD1 3-D model was further evaluated. Antibodies targeting spatially distant peptide sequences are more likely to interact with the native protein without steric hindrance, increasing the likelihood of obtaining antibody pairs for two-site ELISA. Finally, all selected peptides were compared for sequence homology by Basic Local Alignment Search Tool (BLAST) database search to minimize unwanted cross-reactivity to off-target proteins.

The use of Trx as the scaffold carrier exploits its well-known ability to significantly increase the accumulation and solubility of target proteins produced in *E. coli*[42]. All hANKRD1 tripeptide inserts were highly expressed in soluble form, allowing easy production and purification of milligram quantities of the antigens for animal immunization. Additionally, three peptide units are coupled to each Trx molecule (~12 kDa), achieving a higher coupling ratio than what could be achieved with synthetic peptides where 1–1.5 peptide molecules are typically coupled per 10 kDa of carrier protein. Prokaryotic peptide production obviates many difficulties encountered in the synthesis and handling of synthetic peptides in terms of yield, solubility, purity and structural features. The 3-D model of Trx-fused antigens revealed that each unit of the tripeptide insert ostensibly adopts a different structural fold and this may in turn increase the likelihood of the peptides displaying protein-reactive antigenicity. Combining three different tripeptide antigens into one immunogen cocktail effectively presents nine sequences for eliciting protein-reactive antigenicity. Furthermore, using a multi-peptide cocktail

minimizes animal usage, housing and handling costs. DEXT microplates allows small-volume assays (15 μl per well) to be performed for hybridoma clone screening and peptide identification. The flat plate architecture facilitates direct visual selection of positive clones, obviating the need for terminating the enzyme-substrate reaction and absorbance reading. The disposable flat sheets are compact to store and minimizes plastic waste; all these translating to reagent, labour, time, environmental and cost savings.

Peptide-directed antibody generation streamlines the workflow for epitope mapping. Such information is increasingly recognized for its importance in securing and protecting intellectual property of antibodies[43,44]. Epitope mapping typically relies on the known protein antigen sequence as template to generate an array of peptide libraries or in cases where the antigen is unknown, combinatorial and random peptide libraries are used[14,45]. The work of Forsström et al. suggested that linear epitopes are generally confined to five or seven residues[46]. Hence, although the immunizing peptides used here are between 13 and 24 residues, more than one epitope may be present within each sequence. Our method requires substitution of only 51 residues for complete alanine scanning and further detailed mapping by substitutions with other amino acids can be easily made if so desired. The Multipin format of the alanine scan peptide library used allows the critical binding residues of three different mAbs to be simultaneously mapped by one staff in two days. On the other hand, if the whole protein was used for immunization, a library of a hundred or so overlapping peptides covering the entire hANKRD1 sequence must first be generated to identify the general location of the epitope (general scan) for each mAb. This is followed by the generation of another library of shorter overlapping peptides through a series of moving windows of different sizes (window scan) within the identified general scan sequence to define the precise epitope boundaries and finally identification of critical binding residues by amino acid substitution-based analysis.

The availability of antibodies to non-overlapping regions on a protein facilitates antibody validation by the method of 'independent antibody' assessment[47]. In immunostaining experiments, mAbs have been observed to produce identical staining patterns in tissues from wild-type and knockout mice[48,49]. Hence, mAbs targeting different sites on the same molecule are useful for antibody validation where the same staining pattern or their co-localization in double label studies provide important evidence of antibody specificity[50]. Such antibody panels are also useful for investigating regulatory mechanisms of protein function mediated by gene expression modulation, proteolytic processing, protein–protein interactions and subcellular localization. As exemplified by our mAb panel, while immunostaining with AG1/mAb 30D12 and AG5/mAb 4F2 consistently showed upregulation of rANKRD1 in a rat MI model, the observed cellular localization of this protein differed between the two antibodies. Nuclear and cytoplasmic detection of rANKRD1 in rat heart tissue by AG1/mAb 30D12 compares well with the nuclear and sarcomeric I band localization observed in fetal and adult rat cardiac myocytes using rabbit polyclonal antibodies directed at the unique N-terminal domain of ANKRD1[32,33]. However, probing with C-terminal-specific AG5/mAb 4F2 and HPA038736 polyclonal antibody reveals corroborative presence of rANKRD1 in the cytoplasm alone. Such differential staining patterns obtained with antibodies recognizing different ends of the same molecule could be indicative that ANKRD1 interacts with DNA or transcription factors via its C-terminus ankyrin units, thus abolishing AG5/mAb 4F2 binding in the nucleus. The availability of mAbs targeting both terminal regions of ANKRD1 will also be highly useful for delineating the implications of proteolytic processing of this cardiac-specific protein in health and disease.

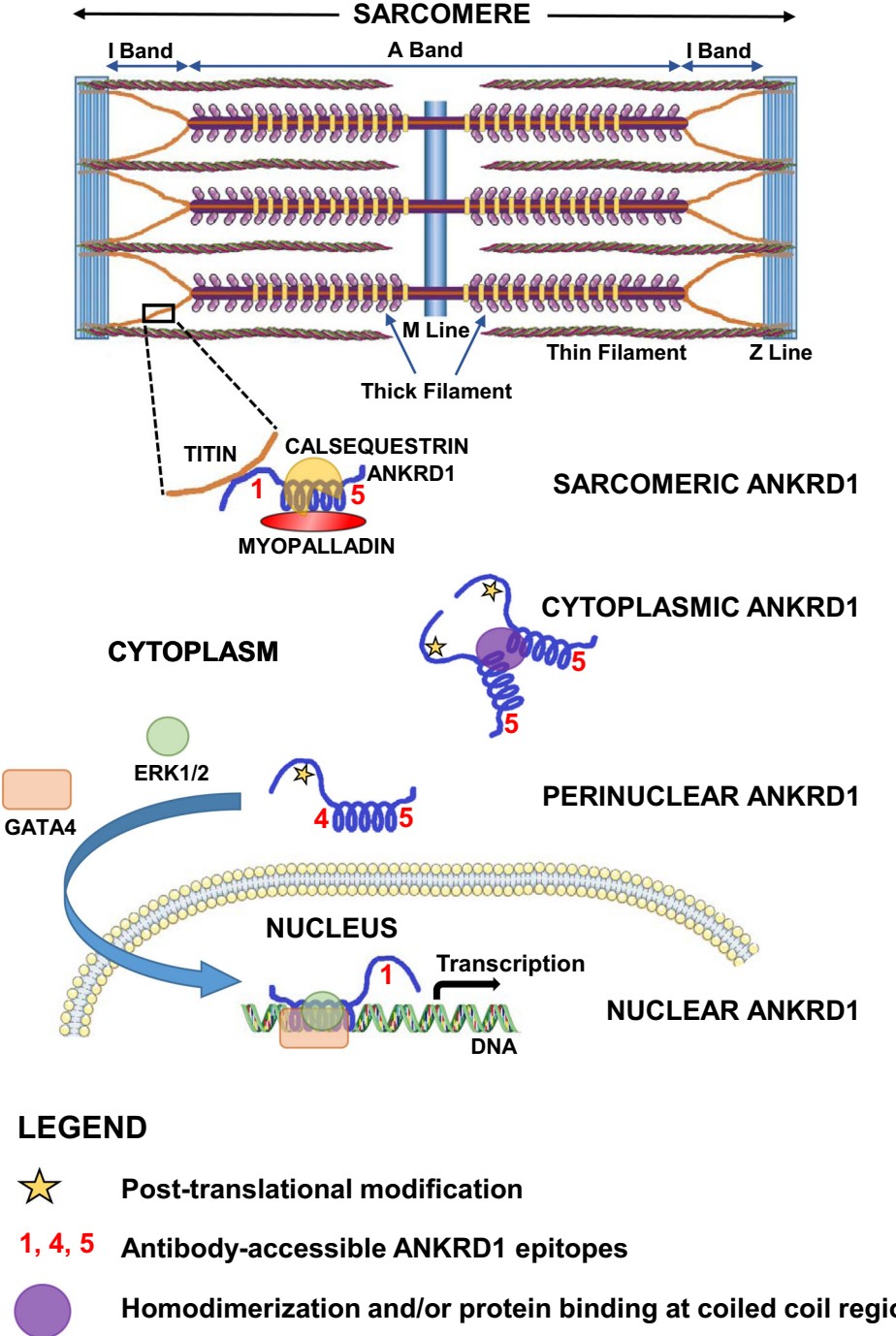

**Fig. 8 Proposed mechanistic scheme depicting ANKRD1 cellular localization.** In cardiac myocytes, ANKRD1 is localized within the sarcomere and is associated with titin, myopalladin and possibly calsequestrin. In H9c2 cells, the AG4 epitope in cytoplasmic ANKRD1 is inaccessible possibly due to steric hindrance by homodimerization or protein interaction at the coiled coil domain. The AG1 epitope with potential post-translational modification, possibly phosphorylation at the epitope critical binding residue Y33, abolishes antibody recognition by AG1/mAb 30D12. Putative interaction of C-terminal ankyrin units with nuclear components blocks AG4/mAb 5C1 and AG5/mAb 4F2 binding in the nucleus. Template elements available on Servier Medical Art licensed under a Creative Commons Attribution 3.0 Unported License (https://smart.servier.com) and art tools from Microsoft Paint were used to generate the figure schematics.

The availability of mAbs targeting multiple sites on ANKRD1 can also facilitate a mechanistic understanding of its subcellular localization as illustrated in Fig. 8. Although H9c2 cells lack sarcomeric organization and do not fully reflect the functional and structural physiology of ventricular cardiomyocytes[51], the nuclear and cytoplasmic staining patterns of the mAbs in these cells largely mirrored that observed in rat heart tissue sections.

The primary difference relates to weak perinuclear localization detected with AG4/mAb 5C1 in H9c2 cells while complete lack of staining was observed in rat cardiac cells. The latter observation may be due to ANKRD1 interaction with sarcomeric proteins like myopalladin and calsequestrin at the coiled coil region in the vicinity of the AG4 epitope. We propose that endogenous rANKRD1 in the cytoplasm of H9c2 cells may exist as

homodimers or be bound to some interacting proteins at the coiled coil domain to render the AG4 epitope inaccessible for antibody binding. It is possible that at the perinuclear region, tight binding at the coiled coil region may be somewhat relaxed to expose the AG4 epitope for limited antibody binding. This may be mediated either by disruption of ANKRD1 dimers to its monomeric form or loss of protein binding interactions at the coiled coil region. ANKRD1 contains two putative N-terminal nuclear localization motifs and the "relaxed" coiled coil domain may facilitate ANKRD1 recruitment of GATA4 and ERK1/2 to form a macromolecular complex that mediates translocation to the adjacent nucleus to effect transcriptional gene regulation[52]. It is postulated that the C-terminal ankyrin units are involved in binding interaction with nuclear components as revealed by the strong detection of a 250-kDa band with AG1/mAb 30D12 but not AG5/mAb 4F2 and weak detection with AG4/mAb 5C1. Intriguingly, Western blot data showed that AG1/mAb 30D12 failed to detect endogenous full-length rANKRD1 in H9c2 cell lysate although its presence may be indicated by the positive 40-kDa bands obtained with AG4/mAb 5C1 and AG5/mAb 4F2. The NetPhos 3.1 server[53] (http://www.cbs.dtu.dk/services/NetPhos/) predicts phosphorylation by an unspecified kinase at position Y33 (probability score of 0.869), shown by alanine scanning to be a critical residue for AG1/mAb 30D12 recognition. We surmise that post-translational modification, possibly Y33 phosphorylation, abolished AG1/mAb 30D12 binding to endogenous rANKRD1 while vector-expressed hANKRD1 lacking this post-translational modification at the AG1 epitope could be recognized by the antibody.

Antibodies targeting the terminal regions of hANKRD1 (AG1 and AG5) displayed broader utility in downstream applications compared with those that recognize an internal peptide sequence (AG4). Terminal regions of proteins are known to be more surface exposed and flexible compared with internal domains[54], and this could allow more favorable energetic consequences for antigen–antibody interactions and conformational accessibility to AG1 and AG5 mAbs. Self-pairing mAbs (AG1/mAb 30D12 and AG5/mAb 4F2) could be used as matched pairs in two-site immunoassays and exploited to measure levels of dimeric and monomeric forms of circulating ANKRD1. In contrast with observations of upregulation in infarcted heart tissue, the use of this matched antibody pair showed downregulation of plasma rANKRD1 in post-MI day 7 with recovery towards normal levels by day 30. These antibodies offer the tantalizing prospect of conducting time course measurements of circulating ANKRD1 and evaluating their diagnostic/prognostic value as early markers of myocardial injury and in predicting outcome.

We previously reported a similar surface presentation strategy by inserting antigenic peptides (9–19 amino acids long) into the active site of Trx that led to breakthrough success in generating mAbs specific to three common single amino acid mutants (R175H, R248Q and R273H) of tumor suppressor p53[16]. All mAbs generated detected their intended mutant proteins without cross-reacting with other mutant p53 proteins or to the wild-type p53 protein. Demonstration that the R248Q mAb was unable to detect transfected or endogenous R248W mutant protein by direct immunoblotting, immunoprecipitation and immuno-fluorescence methods provided unequivocal evidence of the specificity and practical usefulness of the mAbs generated. In this early work, single antigen immunizations were performed and antigen peptide selection was dictated by the location of the mutation hotspot on the p53 sequence. Notably, all recombinant p53 mutant tripeptide fusion proteins produced in E.coli were primarily expressed in insoluble inclusion bodies, which necessitated protein refolding to achieve solubility of the antigens in physiological buffer suitable for immunization. Taken together,

our method is robust even in cases where peptide antigens cannot be selected on the basis of its predicted antigenicity or physico-chemical properties but the inherent drawback is that of increased likelihood of insoluble heterologous expression and additional downstream processing for immunogen preparation.

We described an integrated resource-conserving workflow that addressed current problems with antibody generation, validation and quality. Protein-reactive anti-peptide mAbs were elicited at high frequency and worked well under native and denaturing conditions. Primary data were provided for antibody character-ization and application-specific validation but further detailed work is warranted to enhance the level of validation in accord with the context of the required experimental application[55]. This streamlined approach for producing quality antibody panels will be of wide interest to academia and industry.

## Methods

**Immunogen design and generation**. We analyzed the protein sequence of human ANKRD1 (GenBank Accession No. EAW50116.1) using the B-Cell Linear Epitope Prediction tool on the Immune Epitope Database Analysis Resource server (http://www.iedb.org) and found five predicted epitopes ranging between 12 and 24 amino acids long. Three putative antigenic sequences, TGKKNGNGEAGEFLPEDF RDGEYE (residues 11–34; designated AG1), LSDKNNPDVCDEY (residues 140–152; designated AG4) and DLNAKDREGDTPLH (residues 244–257; desig-nated AG5), were selected for fusion as independent three-copy inserts into the surface exposed active loop of thioredoxin (Trx) harbouring a C-terminal poly-histidine tag. The amino acid sequence of the protein immunogens and their 3D structure are predicted on the basis of homology modeling against thioredoxin 1 (1oaz.1.A.) as template using SWISS-MODEL (http://swiss.model.expasy.org/)[56–59].

The fusion proteins were expressed in E. coli host strain BL21 (DE3) trxB (Novagen, Merck-Millipore, Darmstadt, Germany). Bacterial cultures were grown at 37 °C under ampicillin selection (100 µg/ml) in 1-L shake flasks containing 200 ml of Terrific Broth (Conda Pronadisa, Spain). When the culture reached an $OD_{590}$ of 0.8, isopropyl-β-D-thiogalactopyranoside (IPTG) was added to a final concentration of 0.5 mM to induce fusion protein expression. Cells from 4-h post-induced cultures were lysed using BugBuster® HT Protein Extraction Reagent (Merck-Millipore) and the soluble lysate was recovered by centrifugation at 8000 × g for 20 min at 4 °C[60,61]. Soluble fusion proteins were purified by automated affinity chromatography using 1-ml bed volume Mini Profinity™ IMAC cartridges on the Profinia Protein Purification System (Bio-Rad, Hercules, CA) under the default program settings of the Native IMAC method with desalting[62]. His-tag bound fusion proteins were eluted with 4 ml Elution Buffer (250 mM imidazole, 500 mM NaCl, 100 mM sodium phosphate, pH 8.0) and desalted through a Bio-Gel® P-6 cartridge into 100 mM sodium phosphate, pH 7.2.

**Mouse immunization and hybridoma generation**. A cocktail comprising equi-molar concentrations of Trx-AG1, -AG4 and -AG5 (final total concentration 1 mg/ml) was added to an equal volume of Sigma Adjuvant System (Sigma–Aldrich) and mixed to homogeneity. This cocktail-adjuvant emulsion (total volume 0.1 ml per site) was injected into five 8-week old Balb/c female mice (InVivos, Singapore). The immuni-zations were performed intra-peritoneally on Day 0, 21 and 63, and subcutaneously on Day 42 and 84 (Biological Resource Centre, A*STAR, Singapore). Ten days after the fifth immunization, blood was taken from the tail vein of each mouse and processed to obtain serum. This was then tested against Trx-AG1, -AG4 and -AG5 by ELISA. Two mice with the highest serum antibody titer were selected as the spleen donors for fusion on Day 108, 3 days after a final intravenous injection of the antigen cocktail without adjuvant was given on Day 105.

Mouse myeloma SP2/0-Ag14 cell line (ATCC® CRL-1581™) was used as the fusion partner, and was cultured in ATCC formulated Dulbecco's Modified Eagle's Medium supplemented with 10% fetal bovine serum (HyClone). Cells were maintained at a density of $5 \times 10^4$–$4 \times 10^5$ cells/ml (early-mid log phase) and were adapted to ClonaCell®-HY Pre-Fusion Medium A (StemCell Technologies, Canada) at least 1 week prior to fusion. Fusion of immune spleen cells with mouse SP2-0/Ag14 myeloma cells using the ClonaCell-HY hybridoma kit was carried out according to manufacturer's instructions (StemCell Technologies, Canada). Briefly, for each mouse spleen, $2 \times 10^7$ parental myeloma cells and $1 \times 10^8$ viable splenocytes were fused in the presence of the fusing agent polyethylene glycol and then allowed to recover for 24 h in recovery medium. Thereafter, the fused cell suspension was mixed with medium containing methylcellulose and plated in ten 100 mm petri dishes. After 10–14 days, single colonies visible to the naked eye were individually picked from the semi-solid media and transferred into single wells of 96-well culture plates. A new sterile tip was used for each colony to maintain clonality of the colony. Plates were incubated for 3–4 days without feeding. Supernatant (100 µl) containing secreted mAbs were then collected and tested by ELISA on DEXT microplates. All approved animal experiments were performed in

compliance with A*STAR Institutional Animal Care and Use Committee (IACUC) regulations.

**DEXT microplate performance validation**. The flat sheets of DEXT microplates [Dextech (S) Pte Ltd, www.dextechlab.com] are made of amorphous polyethylene terephthalate (APET). They are housed in a custom-made base to allow microplate transfer without accidental contact with the samples. Arrayed wells of 6 mm diameter were positioned on the flat sheet in compliance with the dimensional footprint and tolerance specifications for standard microplates outlined by the American National Standards Institute (ANSI) and Society for Laboratory Automation and Screening (SLAS), thus allowing full operability of the DEXT microplates across broad application platforms. Full coverage of each scribed well requires liquid volume of at least 10 μl. Well edges are able to provide sufficient pinning action on liquid drops up to 15 μl without dislodgement even when the DEXT microplate is shaken up to 700 rpm (3 mm orbit).

Well-to-well consistency of the DEXT microplate was assessed by determining whole-plate ELISA response profiles and compared against the performance of two commercially available standard microplates, clear high-binding microplates from Corning (Costar #2592) and Nunc (Maxisorp #439454). The performance of each plate type was tested at three antigen coating concentrations and seven independent assays for each coating concentration were performed by 1 operator over a 1 year period using different batches of reagents.

We previously used a sandwich ELISA measuring human N-terminal pro-Urocortin 2 (NT-proUcn2) to characterize microplate well-to-well variability in immunoassay readout[63]. Here, we use the same antigen–antibody pair to establish the performance of the DEXT microplates. Wells were coated with thioredoxin-fused NT-proUcn2 (15 μl per well) at 0.1, 1 or 10 μg/ml concentrations in 50 mM carbonate-bicarbonate buffer (pH 9.6) and incubated overnight at 4 °C in a humidified enclosure. Antigen drops were removed by inverting the DEXT microplate over absorbent towels. The plates were washed by serial immersion into phosphate buffer (300 ml) containing 0.05% Tween 20 (PBST) with gentle shaking and flicked dry after the third wash. Well surfaces were blocked for 2 h at room temperature with 15 μl of 1% BSA at 4 °C. BSA drops were removed by absorbent towels and the microplates washed as described above. Fifteen μl of anti-NT-proUcn2 mouse monoclonal antibody was added to each well. After incubation for 1 h at room temperature, the same wash procedure was performed and 15 μl of goat anti-mouse immunoglobulin conjugated to horse radish peroxidase (HRP) diluted 1000× with PBST/1%BSA was added. The microplates were then incubated for 1 h at room temperature. After three washes with PBST, 10 μl of 3,3′,5,5′-Tetramethylbenzidine (TMB) was added as substrate to each well for colorimetric detection. The reaction was stopped after 10 minutes with 5 μl of 2 N sulphuric acid. Absorbance was measured on the Enspire Multimode Microplate Reader (Perkin Elmer) at 450 nm with background subtraction at 570 nm. For the conventional 96-well microplates, the same assay procedure was used except that antigen and antibodies were added at 100 μl per well while blocking agent and washes were performed at 300 μl per well. Colour development was achieved by addition of 100 μl of TMB and the reaction stopped with 50 μl of 2 N sulphuric acid. Background subtracted absorbance (OD) values were plotted against well positions to reveal any systematic trends.

**DEXT microplate-based hybridoma screening**. Antigens Trx-AG1, -AG4 or -AG5 at 1 μg/ml were added to the wells of the DEXT microplate (15 μl per well) and incubated overnight at 4 °C in a moisture saturated enclosure. After washing unbound antigen and blocking the wells with BSA, 15 μl of culture supernatant from each hybridoma clone was added to wells coated separately with the three antigens. Subsequent ELISA procedures were as described except that the addition of stop solution and absorbance measurements were omitted. Hybridoma clones were selected when a strong intense blue colour was visually observed in only 1 of the three antigen wells. Selected positive clones were subcultured and retested by ELISA against all three antigens and the thioredoxin carrier protein. After second round screening, positive clones with high absorbance specific to one of the four test antigens (Trx, Trx-AG1, Trx-AG4 and Trx-AG5) were selected for subcloning using limiting dilution and subjected to further cell expansion, antibody purification and characterization.

**Antibody purification and Isotyping**. The isotype of each mAb was determined using the Pierce^TM Rapid ELISA Mouse mAb Isotyping Kit according to the manufacturer's instructions. IgG mAbs were purified from 100 ml of high-density hybridoma culture supernatant (7–10 days stationary batch) by automated affinity chromatography using 1 ml bed volume Mini Affi-Prep® Protein A cartridges. Briefly, hybridoma culture supernatants were first concentrated 10-fold using centrifugal protein concentrators (Sartorius, Germany), then diluted with Pierce^TM Protein A IgG binding buffer (1:4 ratio) and filtered through a 0.22-μm filter before being purified on the Profinia^TM Protein Purification System (Bio-Rad, USA) under the default program settings of the Protein A/G affinity method. The mAbs were eluted using Pierce^TM IgG elution buffer. In the case of IgM mAbs, hybridoma culture supernatant was purified using HiTrap IgM purification HP column according to the manufacturer's instructions (GE Life Sciences, USA).

Concentration of the purified mAbs was determined using the Pierce^TM BCA Protein Assay Kit according to the manufacturer's instructions. The purity of the mAbs was assessed by 16% SDS-polyacrylamide gel analysis.

**Antibody characterization by surface plasmon resonance**. Determination of the association ($k_a$), dissociation ($k_d$) and equilibrium (KD) constants of mAbs against full-length ANKRD1 was performed on the ProteOn XPR36 (Bio-Rad, Hercules, USA) at 25 °C on GLC chips. The sensor surfaces were activated with a mixture of 4 mM N-(3-dimethylaminopropyl)-N-ethylcarbodiimide hydrochloride (EDAC) and 1 mM N-hydroxysulfosuccinimide (NHS) in the vertical orientation. Purified mAbs and bovine serum albumin as control were diluted to 10 μg/ml in 10 mM sodium acetate buffer (pH 5.0) and immobilized to about 3000–5000 response units (RU). Inactivation of residual activated carboxyl groups on the chip surface was achieved by treatment with 1 M ethanolamine-HCl (pH 8.5). Recombinant hANKRD1 (N-terminal His-tagged; OriGene Technologies, USA) in PBST (phosphate buffer saline containing 0.05% Tween 20) between 1.1 and 270 nM was injected in the horizontal orientation for 180 s at a flow rate of 100 μl/min and dissociation was monitored for 30 min. The sensorgrams were processed using the "Auto Process" function on the ProteOn Manager software to sequentially perform injection and baseline alignment as well as artifact removal. For kinetics determination, double subtraction of the processed sensorgrams from both the 'PBST' buffer and 'BSA' channel reference was performed. The binding curves were fitted against a 1:1 Langmuir binding model (with drift) to obtain the grouped association (ka), dissociation (kd) and equilibrium constants (KD) derived from at least three selected analyte concentrations that gives Chi$^2$ values less than 10% of R$_{max}$ (maximum response when all available ligand is occupied).

After interaction with full-length hANKRD1, the sensor surface was regenerated using 0.85% phosphoric acid at a flow rate of 100 μl/min for 18 seconds followed by PBST injection at 100 μl/min for 60 seconds. The appropriate recombinant thioredoxin-fused antigen diluted in PBST to concentrations between 3.33 and 270 nM was then injected in the horizontal orientation to evaluate the binding interaction of the respective mAbs with their corresponding antigen.

**Alanine scanning**. Peptides (Multipin system)[64] corresponding to the sequences of AG1, AG4 and AG5 but with one alanine substitution of individual amino acids, were used to determine critical residues for antibody–antigen binding (Mimotopes, Australia). Test procedures were carried out according to the manufacturer's instructions. Briefly, the gears with bound peptides were pre-coated with PBS containing 0.1% Tween 20 for 1 h at room temperature to block nonspecific absorption of antibodies. Overnight incubation at 4 °C with each mAb (1 μg/ml) diluted in 1% BSA/PBS was followed by a PBS wash for 10 min. Reaction for 1 h with a goat anti-mouse antibody conjugated with HRP (1:3000 dilution in 1% BSA/PBS) was again followed by another PBS wash for 10 min. The presence of antibody was detected by reaction with TMB (Sigma–Aldrich) for 10 min and colorimetric signals read on the Enspire Microplate Reader (Perkin Elmer) at 650 nm. Prior to retesting, the bound antibody was removed from the peptides by immersing the peptides in a sonication bath (Branson 200) containing 0.1 M phosphate buffer/1% SDS/0.1% 2-mercaptoethanol for 10 min at 60 °C before washing several times with purified water. Complete removal of bound antibodies was confirmed by conjugate testing according to manufacturer's instruction.

**Utility of mAbs**. The utility of the mAbs were tested for applications in ELISA, western blotting and immunohistochemistry. Antibody pairs are identified by ELISA using a checkerboard screening method. IgG mAbs were coated onto microplate wells at 5 μg/ml and unbound sites blocked with 3% BSA in phosphate buffer. Recombinant hANKRD1 (20 ng/ml) was added to the wells for binding with the coated antibodies. Unbound proteins were washed off and biotinylated mAbs (1 μg/ml) were added to each well. Signal was generated by the action of streptavidin-HRP on TMB substrate and absorbance read at 450 nm. Recombinant hANKRD1 was omitted for the blank wells of each antibody pairing. Antibody pairs that gave strong positive signals were selected for further evaluation. With capture and detection antibody concentrations kept constant, recombinant hANKRD1 was titrated to generate a 7-point standard curve. Absorbance values were plotted against hANKRD1 concentrations and fitted against a linear regression model.

**Two-site sandwich ELISA**. The antibody pair with the best signal-to-noise ratio was used to develop a two-site sandwich ELISA test for hANKRD1. mAb 4F2 (5 μg/ml in 50 mM carbonate-bicarbonate buffer, pH 9.6) was used as the capture antibody and mAb 30D12 (1 μg/ml) was used as the detector antibody. Recombinant hANKRD1 was used as the calibrator. Briefly, 100 μl of mAb 4F2 was added to each well of high-binding polystyrene microplates (Costar®, Corning) and incubated overnight at 4 °C. Unbound surfaces were blocked with 3% BSA in PBS for 2 h at room temperature. Calibration curves were generated with 7 calibrators at 20, 8, 3.2, 1.28, 0.512, 0.2048, 0.082 ng/ml (diluted in 2.7% BSA-PBS) of hANKRD1. Rat blood was collected from the left ventricle upon euthanasia at day-7 and day-30 post-MI. EDTA plasma was prepared after 4000 rpm centrifugation for 10 min at 4 °C, aliquoted and stored at −80 °C until assay. Each plasma sample was pre-diluted 3-fold with PBS containing

100 μg/ml of Super ChemiBlock (Merck-Millipore). Calibrators, quality controls and test samples were added in duplicates to the antibody-coated microplate (100 μl/well) and incubated for 2 h at room temperature with shaking at 500 rpm. After four washes with PBST, biotinylated mAb 30D12 was added into the wells and incubated for 1 h at room temperature with shaking at 500 rpm. Wells were again washed four times and streptavidin-HRP was added followed by incubation for 30 min at room temperature with shaking at 500 rpm. After four PBST washes, 100 μl of TMB substrate (Sigma–Aldrich) was added. Reaction was stopped with 2 N sulphuric acid after 30 min and colorimetric signals were read on the Enspire Microplate Reader (PerkinElmer). Results were interpolated from a 5-parameter logistic model and applied to the calibrators using the Enspire® software.

**Western blot analysis**. Recombinant hANKRD1 and homogenates from rat cardiac tissues and neonatal rat cardiomyocytes were used for western blot analysis. A positive control rabbit pAb raised against the C-terminal 56-residue fragment of hANKRD1 (Atlas Antibodies, Sweden, product number HPA038736, lot number A106040) was used for comparing immunoreactivity to the same samples. Protein samples were separated on 12% SDS-PAGE and electroblotted onto an Immuno-Blot® PVDF membrane (BioRad Laboratories). Detection of hANKRD1 was performed using the purified mAbs (0.5 μg/ml) and a goat anti-mouse secondary antibody conjugated with horse radish peroxidase (diluted 1:2000; Agilent Technologies, USA). Protein bands were visualized using a chemiluminescence substrate (Thermo Fisher Scientific, USA).

**Antibody validation by overexpression in H9c2 cell line**. Full-length hANKRD1 cDNA was synthesized and cloned into pJ404 (ATUM, Newark, CA, USA) and pCMV-Myc (Clontech, CA, USA) for bacterial and eukaryotic expression, respectively. Authenticity of the cDNA and its in-frame fusion downstream of the myc tag was verified by DNA sequencing. hANKRD1 was expressed in *E. coli* strain BL21 (DE3) *trx*B as described above. pCMV-Myc and pCMV-Myc/hANKRD1 plasmids were transfected into H9c2 rat myoblast cells using Lipofectamine 2000 (Invitrogen, CA, USA). Transfected and non-transfected cells were lysed in 150 mM NaCl, 50 mM Tris-Cl (pH 7.4), 1% IGEPAL CA-630, 2 mM EDTA containing protease inhibitor cocktail (Sigma–Aldrich P8340). The amount of ANKRD1 present in soluble bacterial and H9c2 lysates (2 mg/ml) diluted at 10, 20, 40 and 80 folds was determined by ELISA as described above. Peptide competition was performed by blocking the detection antibody (AG1/mAb 30D12) overnight at 4 °C with matched (AG1) and unmatched (AG4) peptides at 100 μg/ml. Dilutional linearity and parallelism of endogenous rANKRD1 and transfection-derived hANKRD1 were assessed. Western blot analysis of *E. coli* and H9c2 lysates was also performed. Anti-myc rabbit polyclonal antibodies were from Sigma–Aldrich (CS3956). Immunostaining of H9c2 cells with AG1/mAb 30D12 (10 μg/ml), AG4/mAb 5C1 (20 μg/ml) and AG5/mAb 4F2 (20 μg/ml) was detected by Alexa 488 anti-mouse secondary antibody and counterstained with DAPI. Blocking of the anti-hANKRD1 mAbs with matched and unmatched peptides was also performed.

**Immunoprecipitation-mass spectrometry (IP-MS) antibody verification**. Immunoprecipitation from hANKRD1-expressing *E. coli*, neonatal rat cardiomyocytes and pCMV-Myc/hANKRD1-transfected H9c2 rat myoblast cell lysates were performed using biotinylated AG5/mAb 4F2. Total protein in each sample was 14.1, 1.13 and 1.32 mg, respectively. Plasma was pre-cleared of endogenous biotin by incubating 100 μl aliquots overnight at 4 °C in neutravidin-coated microplates. Capture antibody was added to the cell lysates and recovered by immobilization onto streptavidin magnetic beads (Dynabead M-280 Streptavidin). The beads were washed three times each with PBST and PBS. Bound proteins were eluted in 1X SDS protein loading buffer containing 1% beta-mercaptoethanol.

Tryptic digestion was performed on the immunoprecipitation eluates using the S-Trap column (Protifi). The digested peptides were vacuum-dried and reconstituted in 2% acetonitrile, 0.1% formic acid for MS analysis. Online reversed-phase (RP) separation of the reconstituted peptides was first performed on the nanoLC Ultra system (Eksigent). The RP solvent A was 2% acetonitrile, 0.1% formic acid, while solvent B was 98% acetonitrile, 0.1% formic acid. The peptides were first trapped on a ProteoCol C18P precolumn (3 μm 120 Å, 300 μm × 10 mm; Trajan), and then resolved on a ChromXP C18-CL analytical column (3 μm 120 Å, 75 μm × 150 mm; Eksigent). Peptide separation was achieved using a two-step linear gradient of 5–12% solvent B over 20 min and 12–30% solvent B over 40 min. Eluted peptides were injected directly into the SCIEX TripleTOF 5600 system operating in information-dependent acquisition (IDA) mode. Peptide precursors were acquired across a mass range of 350–1250 *m/z* with 250 msec accumulation time. Selection of maximum 50 precursors was allowed per cycle. MSMS fragment ion information was acquired across 100–1800 *m/z* with 50 msec accumulation time, in high sensitivity mode and with dynamic exclusion of 15 sec and rolling collision energy enabled. Acquired spectra were searched using the ProteinPilot 5.0 software (SCIEX) against either (i) the SwissProt *Escherichia coli* pan-strain K12 reference proteome (2020 July release, 4389 entries) spiked with common contaminant proteins and the hANKRD1 sequence (sp|Q15327|ANKR1_Human); or (ii) the UniProt *Rattus norvegicus* reference proteome (2020 June release, 29,242 entries) spiked with common contaminant proteins and the hANKRD1 sequence.

The 'thorough search' mode of the Paragon search engine (v5.0.0.0) was used, with the following parameters specified: cysteine alkylation with methyl methanethiosulfonate, common biological modifications and detected protein threshold at 0.05. False discovery rate (FDR) analysis was performed against decoy reversed sequences generated from the respective protein databases. The threshold for determining confident protein identifications was set at a 95% confidence cut-off (corresponding to <1% FDR) and at least one peptide identified.

**Immunohistochemistry**. The in vivo rat model of myocardial infarction (MI) by ligation of the left anterior descending (LAD) coronary artery was applied. Male Sprague-Dawley rats, weighing 170–220 g, were used. The animals were kept in a temperature-controlled room (21 ± 2 °C) with 12 h light and dark cycle. Water and diet were available ad libitum. Animals were anesthetized with ketamine and xylazine (75 and 10 mg/kg, respectively, i.p.), intubated and ventilated. The chest on the left side was opened by an incision parallel to the ribs at the fourth intercostal space. The LAD coronary artery was ligated using 7/0 polypropylene suture. The chest was closed. An identical surgical procedure without LAD ligation was carried out for the sham control group (Sham).

For immunostaining, rat heart tissue from day-7 post-MI was fixed in neutral buffered formalin, processed and embedded into paraffin blocks. Transverse sections (5-μm thick) were cut from the mid-section of the ventricles. Antigen retrieval was performed by heat treatment in citrate buffer (Sigma–Aldrich). After blocking with 5% goat serum, primary antibodies (mAbs 30D12, 4F2 or 5C1; 10 μg/ml) and biotin-conjugated wheat germ agglutinin (WGA) (Vector Laboratories, Inc., Singapore) were incubated at 4 °C overnight, followed by secondary antibody (Goat anti-mouse IgG-Alexa Fluor 594), streptavidin-Alexa Fluor 488, and 4′,6-Diamidino-2-Phenylindole, Dihydrochloride (DAPI) (Invitrogen, Thermofisher, Singapore) at room temperature for 1 h. Immunofluorescence was visualized using the Nikon Eclipse Ti inverted microscope with C-HGFIE pre-centered fiber illuminator (Nikon, Singapore) and appropriate filter sets.

**Statistics and reproducibility**

*GraphPad QuickCalcs*. *t*-test calculator was used for group comparisons of experimental myocardial infarction (MI) model rats. Two-tailed unpaired *t*-test was used to compare each two-group combination with significance at $p < 0.05$. The number of independent animals used for each group are described in the legend to Fig. 4. With limited plasma availability from each animal, measurement of rat plasma ANKRD1 was performed in duplicates in one ELISA assay. Intra-assay CV of all plasma ANKRD1 reported are less than 12%.

**Reporting summary**. Further information on research design is available in the Nature Research Reporting Summary linked to this article.

## Data availability

All data are available from the corresponding author upon reasonable request. Raw data for (Figs. 3b, d, 4a–d) and mass spectrometry of antibody pull-down from *E. coli* expressing human ANKRD1 are available in Supplementary Data 1 and 2, respectively.

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

## Acknowledgements

Funding support for this work was provided under the Centre Grant awarded to A.M.R. by the National Medical Research Council, Singapore.

## Author contributions

O.W.L. and S.L.S.M. designed and developed the antibody production methodologies, performed most of the experiments, analyzed the data and wrote the paper; S.L. and E.M. W.N. validated DEXT microplate assay performance and liquid drop evaporation, respectively; S.L., J.P.C.C., Y.X.N., A.E.S.L. and E.R.Y.L. performed hybridoma screening and analysis of results; Y.Z. and P.P.W. performed the rat MI and H9c2 transfection

experiments as well as immunofluorescence staining work; Q.F.L., Q.S.L. and T.K.L. performed the tryptic digest and mass spectrometry analysis of immunoprecipitated eluates for antibody verification; T.W.N. conceived and developed the DEXT microplates; A.M.R. contributed key intellectual inputs to the manuscript and is the principal investigator of the grant that supported the projected. All authors have contributed, discussed and approved the final version of the manuscript.

## Competing interests
The authors declare no competing interests.
