## [Peer Review File · Communications Biology]

Reviewers' comments:

Reviewer #1 (Remarks to the Author):

General comment:

The present study developed an epitope-directed approach to generate anti-peptide mAbs targeting several nonoverlapping protein sites by epitope prediction. The authors did lots of jobs to conduct this research. However, the innovation of the study is not significant and there have already many reports about antibody preparation using a portion of the peptide of the target protein. Meanwhile, the effectiveness of the method needs further demonstration. Overall, this manuscript is not suitable to be accepted in its present form.

Specific comments:

C1

The Introduction is not comprehensive, and the development of "Epitope mapping" and "peptide directed antibody production" should be mentioned.

C2

The normalization of figures in the article is poor. The clarity of the figure is not up to standard and some charts in the manuscript are not necessary.

C3

The results of epitope prediction using protein sequences are often inaccurate and may miss the opportunity to produce high quality antibodies. Therefore, it will be more convincing to compare the results of this study with the whole protein as immunogen.

C4

Why only AG1/mAb 30D12 and AG5/mAb 4F2 displayed dual utility as both capture and detection antibodies whilst other protein-reactive mAbs are useful only as capture antibodies? Is it related to the preparation of antibody with small peptides?

C5

In this study, how many definite recognition sites (1 or 2) on the monoclonal antibody prepared by epitope-directed? Although compared the recognition ability of a series of mAbs to rANKRD1 and hANKRD1 respectively, the evidence is not enough.

Reviewer #2 (Remarks to the Author):

Summary of Paper:

This manuscript describes a method to create multiple monoclonal antibodies to the same protein in one hybridoma production cycle by fusing carefully selected distinct peptides to surface-exposed loops of a thioredoxin carrier. As a proof-of-principle, the authors developed antibodies against hANKRD1, a protein implicated in dilated cardiomyopathy (DCM) and hypertrophic cardiomyopathy (HCM) with potential to be a cardiac biomarker. Validated antibodies against this protein would enable mechanistic studies and also have potential diagnostic and prognostic ability. Hybridoma supernatants arising from this method were rapidly screened using an ELISA assay on novel DEXT microplates. Positive clones underwent affinity measurements, isotyping and epitope mapping as part of their characterization before being tested by two-site ELISA, western blotting and immunohistochemistry. The method yielded 24 stable hybridoma clones, 15 antibody pairs for two-site ELISA, and 9 clones positive for western blotting illustrating the effectiveness of this method for hANKRD1.

Overall Impression:

The authors have developed a streamlined method for antibody production that when successful would

produce well-characterized antibodies for downstream applications. The multiplexing at the hybridoma stage would significantly reduce time and labour costs and the DEXT microplates enable rapid screening of the hybridomas. Importantly, this method facilitates development of multiple distinct antibodies that can provide further confidence in antibody specificity and enable assays such as two-site ELISAs. The manuscript was well written, and it will be of interest to those interested in developing antibodies or more generally to the biological community as an example of how antibodies could be developed and characterized.

Major Comments:

1. My main concern is that the authors advance this as a new method for antibody development, but they only validate its effectiveness against one protein. The authors should repeat this procedure with at least one to two other proteins to show its general utility.

2. It also was not clear to me if this approach would still be effective against proteins that were less characterized (e.g. no protein structure, lack of knowledge of PTM/protein binding) for which it would be harder to select optimal peptides. This may not necessarily need to be tested but it should at least be discussed.

3. In general, it seemed that the antibodies were used for ANKRD1 characterization in tissue before experiments that are more basic were performed to validate the antibodies. I recommend at least one of three experiments for figure 4: 1) Repeat the two-site ELISA and western blotting in cell lines where ANKRD1 is either overexpressed ectopically or endogenous ANKRD1 is removed by siRNA or CRISPR to show a differential signal, and/or 2) a competition experiment with free peptide to show loss of signal, and/or 3) overexpression of ANKRD1 with mutations in the key residues identified as critical for binding to show loss of signal. This would provide more confidence that they are actually measuring ANKRD1. This is particularly important where the recombinant protein is migrating at a different size on the western blot and their positive control - HPA038736 has not been validated for western blotting (<https://www.proteinatlas.org/ENSG00000148677-ANKRD1/antibody>). I would also recommend that their primary candidate for capture in the ELISA be assessed by immunoprecipitation followed by MS (IP-MS) to confirm that it actually does immunoprecipitate ANKRD1.

4. Similar to point 3, for the IHC in figure 5, I would also recommend that the antibodies first be tested in a more controlled experiment such as in a cell line with ectopic ANKRD1 or depleted ANKRD1 to show a differential signal before being tested in rat tissue. This is particularly important as AG1 showed nuclear staining in contrast to AG5 and the positive control HPA038736. I would also recommend a peptide competition experiment for the IHC to show the signal is specific to ANKRD1.

5. The authors clearly show the advantage of the DEXT microplates for this workflow. They should make it clear how readers could obtain these plates to replicate the workflow.

Minor Comments:

1. Figure 2B – It is not entirely clear how the 14 peptides were selected. My assumption is that these are all those that meet the 0.35 score threshold (coloured yellow in A). Clarifying this would be easier for the reader to understand the progression from Figure 2A to 2B.

2. Line 295 – It is not clear how the authors determined the ELISA assay detection limit as 0.073 ng/ml.

3. Figure 4C – the image is cropped on the right hand side. The HPA antibody is missing the 6 on the end.

4. Line 610 – states “specific to one of the four test antigens” – should this be three (AG1, AG4, AG5)?

5. In Supplementary Figure 4, the font size for the ELISA OD should be increased and I would recommend that they use a different color for the signal of the critical residues to better highlight them.

6. In Supplementary Table 2 – it would be helpful to clarify what the positive and negative control wells are.

Reviewer #3 (Remarks to the Author):

General comments:

The manuscript reads like two separate topics; (1) a description of a hybridoma-production pipeline (based on Figure 1), for generating, characterizing and validating mAbs, and (2) a validation study of several anti-hANKRD1 antibodies using said pipeline. Aside from the mash-up of these two topics, which is a serious issue (see below), the manuscript is well-written and would require little editing.

Because the manuscript is entitled "Efficient epitope-directed mAb production using a mixed Ag cocktail", I will focus my critique on this aspect of the two topics described above.

There is only a single example of the running of the pipeline. So the efficiencies, including the metrics for each step: percent success, successful applied to which applications, costs (and how costs are calculated) etc. are insufficient for anyone to adapt the method. Or even really to understand the demonstrated advantages toward using it. There is little innovation in the methods or pipeline described beyond possibly the use of trimer fusion peptides multiply injected into the animal.

But without a lot more data describing the benefits to speed, quality or cost. The audience for this information would be other users or developers of an Ab pipeline. These would be sophisticated researchers. And the data they would need to evaluate this pipeline is missing.

The work on the isolation and validation of the hANKRD1 is fine. But again, that's not the focus of the paper.

Response to Reviewers

We are pleased to resubmit our revised manuscript entitled “Efficient Epitope-directed Monoclonal Antibody Production Using A Mixed Antigen Cocktail”. We deeply appreciate the helpful suggestions and comments for improvements to our study and have made significant changes to the manuscript (highlighted in yellow in main text for easy tracking). Detailed responses to the constructive critiques and issues raised by each reviewer are addressed below:

Reviewer #1 (Remarks to the Author):

General comment:

The present study developed an epitope-directed approach to generate anti-peptide mAbs targeting several nonoverlapping protein sites by epitope prediction. The authors did lots of jobs to conduct this research. However, the innovation of the study is not significant and there have already many reports about antibody preparation using a portion of the peptide of the target protein. Meanwhile, the effectiveness of the method needs further demonstration. Overall, this manuscript is not suitable to be accepted in its present form.

Specific comments:

C1

The Introduction is not comprehensive, and the development of “Epitope mapping” and “peptide directed antibody production” should be mentioned.

Response: We have amended the Introduction section to provide brief information on peptide directed antibody production along with four additional references (Lines 53-63). We feel that more information on epitope mapping is best provided in the discussion section and have done so along with additional references (Lines 648-650, 658-663).

C2

The normalization of figures in the article is poor. The clarity of the figure is not up to standard and some charts in the manuscript are not necessary.

Response: We have reviewed and improved on the clarity and quality of all the figures in the manuscript.

C3

The results of epitope prediction using protein sequences are often inaccurate and may miss the opportunity to produce high quality antibodies. Therefore, it will be more convincing to compare the results of this study with the whole protein as immunogen.

Response: The focus of this study is to demonstrate that peptide antigens can be used successfully to produce high quality monoclonal antibodies, an approach that facilitates downstream antibody characterization, epitope mapping and validation. As explained in the manuscript, the use of whole proteins as immunogen makes epitope mapping more tedious and laborious and overcoming this challenge with peptide antigens is one of the primary motivation for this work. It is not the objective of this study to compare the efficiency between peptide and protein immunogens but rather that the peptide antigen approach is viable and capable of generating quality protein-reactive antibodies.

We also believe that the use of the whole protein as immunogen will not necessarily lead to production of more high quality antibodies for the following reasons. As a rule of thumb, intact proteins contain one immunogenic site for every 5-10 kDa molecular weight (Sutcliffe et al. 1983, Science, 219, 660-666). A counter-intuitive observation from previous studies show that the immunogenicity of an intact protein is less than the sum of the immunogenicity of its fragments (Green et al. 1982, Cell, 28, 477-487). In other words, peptides have been shown to be able to produce antibodies directed at regions where whole proteins as immunogens could not. Furthermore, it has also been shown that the peptide homologue of the C-helix region of myohemerythrin (MHR) as immunogen was more efficient compared with the intact protein in producing mAbs that are reactive to both MHR and the C-helix peptide (Fieser et al. 1987, PNAS, 84, 8568-8572). In that study, 80% of the anti-peptide hybridomas produced antibodies that were reactive to both the peptide and native protein while only one hybridoma producing protein-reactive antibodies was obtained after three spleen fusions when the whole MHR was used as immunogen.

We extracted the available curated binding affinity data of 87 mouse monoclonal antibodies from the SabDab structural antibody database (Dunbar et al. 2014, Nucleic Acids Research, 42, D1140-D1146) and used the published KD values to classify their binding strength as low (micromolar range; 10^{-5} to 10^{-6} M), moderate (nanomolar range; 10^{-7} to 10^{-9} M), high (picomolar range; 10^{-10} to 10^{-11} M) and very high (10^{-12} M). This data provides an idea of the typical quality of antibodies since the affinity requirement of functional antibodies lies in the nanomolar range. As shown in the pie-chart, 82% of curated antibodies have low to moderate binding affinity.

In contrast, 10 out of 12 peptide antibodies (83%) produced in our work were found to have high binding affinity to the native protein in the picomolar (10^{-10} to 10^{-11} M) range while two mAbs showed binding only to the peptide and not the native protein. This additional information has been included in the discussion section (Lines 533-541). Additionally, our protein-reactive binders can be used in immunoassay, western blotting and cell staining applications. Hence, convincing evidence has been presented in our work demonstrating the ability of the designed peptide antigens to produce a high percentage of quality mAbs that are fit-for-purpose.

C4

Why only AG1/mAb 30D12 and AG5/mAb 4F2 displayed dual utility as both capture and detection antibodies whilst other protein-reactive mAbs are useful only as capture antibodies? Is it related to the preparation of antibody with small peptides?

Response: AG1/mAb 30D12 and AG5/mAb 4F2 recognize the N- and C-terminal regions of ANKRD1, respectively. Terminal regions of proteins are widely known to be more surface exposed and flexible (Jacob and Unger, 2000, *Bioinformatics*, 23(2), e225-e230). Hence, the dual utility of AG1/mAb 30D12 and AG5/mAb 4F2 is likely related to the local conformational accessibility and flexibility of the native protein and possible favourable energetic consequence at these terminal regions, allowing their ready binding as capture and detection antibodies. This explanation with an additional reference has now been included in the manuscript (Lines 666-668).

C5

In this study, how many definite recognition sites (1 or 2) on the monoclonal antibody prepared by epitope- directed? Although compared the recognition ability of a series of mAbs to rANKRD1 and hANKRD1 respectively, the evidence is not enough.

Response: We apologize that we do not quite understand the question asked. We assume the reviewer is asking whether the monoclonal antibody will interact with the native protein via one or both of its antigen binding arms. X-ray crystallographic evidence indicate that a structural epitope, defined as the topographical area of contact between the protein and antibody, involve 15-20 contacting residues and encompass buried surface areas between 650 – 900 Å² for the antibody-antigen interface (Davies et al. 1990, *Ann Rev Biochem*, 59, 449-473). Hence, one binding arm of the epitope-directed mAb is expected to interact with one monomeric ANKRD1 molecule at the cognate antigen site and whether the second Fab arm is occupied by another ANKRD1 molecule will depend largely on the flexibility of the Fab arms and hinge region of the mAb (Roux et al. 1997, *J Immunol.*, 159, 3372-3382) and the potential steric hindrance of the second ANKRD1 by the initially bound molecule (Oda and Azuma, 2000, *Mol. Immunol.*, 37, 1111-1122). Since we show that ANKRD1 can also exist in dimeric form, this adds another level of complexity as to whether one or both antibody binding arms will be occupied by the ligand.

We have performed additional experiments to demonstrate the authenticity of our mAbs. Firstly, we show that AG1/mAb 30D12, AG4/mAb 5C1 and AG5/mAb 4F2 do not recognize the MARP-related family member, ANRKD2, which shares high sequence and functional similarities with ANKRD1 (Supplementary Figure 8). Secondly, we cloned the full-length hANKRD1 cDNA into a bacterial expression vector for protein production in *E. coli*. We also cloned the same cDNA downstream of the myc tag in the eukaryotic expression vector pCMV-Myc. ELISA, western blotting and immunostaining data along with peptide blocking experiments with matched and unmatched peptides further corroborates the specificity of these mAbs (Figures 6 and 7).

We also think that it would be ideal if the method described is tested with more than one protein to demonstrate its generality. We have addressed this issue in the manuscript (Lines 676-691) by elaborating on our previous success whereby the same thioredoxin-tripeptide fusion strategy led to breakthrough generation of mAbs that are specific to single amino acid p53 mutation hotspots

- R175H, R248Q and R273H (Cell Reports, 2018, 22, 299-312). All three mutant-specific mAbs detected their corresponding mutant proteins without cross reacting with other mutant p53 proteins or to the wild-type p53 protein. The fact that the R248Q mAb was unable to detect the R248W mutant protein further highlighted the specificity of the mAb generated. The primary differences in this earlier p53 work compared with the present manuscript are:

- 1) Each mouse was immunized with a single mutant peptide antigen
- 2) There was no scope for epitope prediction in this early work since the peptide antigen used was dictated by the location of the mutation hotspot on the p53 sequence. As such, all the recombinant p53 mutant trimer fusion proteins produced in *E.coli* were primarily expressed in insoluble inclusion bodies which necessitated extra refolding work to obtain soluble proteins in physiological buffer suitable for immunization. Protein refolding is a time consuming and tedious task. It is especially challenging to be able to achieve refolded proteins at high concentrations (>0.5 mg/ml) to minimize animal discomfort during immunization.

Herein lies the benefits of our current work where we show it is possible to combine multiple antigens in a single cocktail for immunization. In addition, where there is more flexibility in antigen peptide selection, the use of epitope prediction guides the selection of peptides not only with higher likelihood of being an effective epitope but also with physicochemical properties associated with soluble bacterial expression as exemplified in our work where all selected antigens were highly expressed primarily as soluble proteins in *E. coli*. All three antigens were easily purified by one-step IMAC to concentrations of at least 1 mg/ml each.

We are confident that our trimer peptide fusion strategy represents a highly effective way of antigen presentation that consistently leads to successful generation of desired mAbs of high specificity and affinity. In fact, since our first report of mAbs against R175H, R248Q and R273H mutations, we have progressed to successfully generate specific mAbs to other p53 mutations; Y220C, G245S, R282W and R248W (unpublished data in collaboration with DP Lane's p53 research group). Hence, our additive improvements to this antigen presentation strategy by introducing epitope prediction, careful peptide selection against background knowledge of protein structure/PTM/protein interaction sites where available, use of a cocktail immunogen and hybridoma screening with novel DEXT microplates described in this current manuscript are geared effectively towards obtaining 'protein-reactive' peptide antibodies and streamlining downstream mAb selection, characterization and application-specific validation.

Reviewer #2 (Remarks to the Author):

Summary of Paper:

This manuscript describes a method to create multiple monoclonal antibodies to the same protein in one hybridoma production cycle by fusing carefully selected distinct peptides to surface-exposed loops of a thioredoxin carrier. As a proof-of-principle, the authors developed antibodies against hANKRD1, a protein implicated in dilated cardiomyopathy (DCM) and hypertrophic cardiomyopathy (HCM) with potential to be a cardiac biomarker. Validated antibodies against this protein would

enable mechanistic studies and also have potential diagnostic and prognostic ability. Hybridoma supernatants arising from this method were rapidly screened using an ELISA assay on novel DEXT microplates. Positive clones underwent affinity measurements, isotyping and epitope mapping as part of their characterization before being tested by two-site ELISA, western blotting and immunohistochemistry. The method yielded 24 stable hybridoma clones, 15 antibody pairs for two-site ELISA, and 9 clones positive for western blotting illustrating the effectiveness of this method for hANKRD1.

Overall Impression:

The authors have developed a streamlined method for antibody production that when successful would produce well-characterized antibodies for downstream applications. The multiplexing at the hybridoma stage would significantly reduce time and labour costs and the DEXT microplates enable rapid screening of the hybridomas. Importantly, this method facilitates development of multiple distinct antibodies that can provide further confidence in antibody specificity and enable assays such as two-site ELISAs. The manuscript was well written, and it will be of interest to those interested in developing antibodies or more generally to the biological community as an example of how antibodies could be developed and characterized.

Major Comments:

1. My main concern is that the authors advance this as a new method for antibody development, but they only validate its effectiveness against one protein. The authors should repeat this procedure with at least one to two other proteins to show its general utility.

Response: We do agree with the reviewer that it would be ideal if the method described is tested with more than one protein to demonstrate its generality. However, this is not a trivial exercise requiring substantial resources/funding and at least 1.5 years to complete. Nevertheless, we have addressed this issue in the manuscript (Lines 676-691) by elaborating on our previous success whereby the same trimer peptide fusion strategy led to breakthrough generation of mAbs that are specific to single amino acid p53 mutation hotspots - R175H, R248Q and R273H (Cell Reports, 2018, 22, 299-312). All three mutant specific mAbs detected their corresponding mutant proteins without cross reacting to other mutant p53 proteins or to the wild-type p53 protein. The fact that the R248Q mAb was unable to detect the R248W mutant protein further highlighted the specificity of the mAb generated. The primary differences in this earlier p53 work compared with the present manuscript are:

- 1) Each mouse was immunized with a single mutant peptide antigen
- 2) There was no scope for epitope prediction in this early work since the peptide antigen used was dictated by the location of the mutation hotspot on the p53 sequence. As such, all the recombinant p53 mutant trimer fusion proteins produced in *E.coli* were primarily expressed in insoluble inclusion bodies which necessitated extra refolding work to obtain soluble proteins in physiological buffer suitable for immunization. Protein refolding is a time consuming and tedious task. It is especially challenging to be able to achieve refolded proteins at high concentrations (>0.5 mg/ml) to minimize animal discomfort during immunization.

Herein lies the benefits of our current work where we show it is possible to combine multiple antigens in a single cocktail for immunization. In addition where there is more flexibility in antigen peptide selection, the use of epitope prediction guides the selection of peptides not only with higher likelihood of being an effective epitope but also with physicochemical properties associated with soluble bacterial expression as exemplified in our work where all selected antigens were highly expressed primarily as soluble proteins in *E. coli*. All three antigens were easily purified by one-step IMAC to concentrations of at least 1 mg/ml each.

We are confident that our trimer peptide fusion strategy represents a highly effective way of antigen presentation that consistently leads to successful generation of desired mAbs of high specificity and affinity. In fact, since our first report of mAbs against R175H, R248Q and R273H mutations, we have progressed successfully to generate specific mAbs to other p53 mutations; Y220C, G245S, R282W and R248W (unpublished data in collaboration with DP Lane's p53 research group). Hence, our additive improvements to this antigen presentation strategy by introducing epitope prediction, careful peptide selection against background knowledge of protein structure/PTM/protein interaction sites where available, use of a cocktail immunogen and hybridoma screening with novel DEXT microplates described in this current manuscript are geared effectively towards obtaining 'protein-reactive' peptide antibodies and streamlining downstream mAb selection, characterization and application-specific validation.

2. It also was not clear to me if this approach would still be effective against proteins that were less characterized (e.g. no protein structure, lack of knowledge of PTM/protein binding) for which it would be harder to select optimal peptides. This may not necessarily need to be tested but it should at least be discussed.

Response: We thank the reviewer for this interesting question. As discussed in the case of p53 mutant peptides, the method is robust even when the antigens are selected not on the basis of its predicted antigenicity or physicochemical properties but rather are restricted by the position of the mutated residue. The consequence is that downstream expression and purification of recombinant antigens could be more difficult and laborious because of the increased likelihood of insoluble expression necessitating protein refolding for immunogen preparation. This additional information has now been added in the discussion section of the text (Lines 683-691).

For poorly characterized proteins where structural and binding interaction sites are not available, we propose that the peptide antigens be designed with a bias towards the terminal regions of proteins which are widely known to be more surface exposed and flexible (Jacob and Unger, 2000, *Bioinformatics*, 23(2), e225-e230). However, consideration has to be given to evaluating the true N-termini of mature proteins in cases where proteins are produced with a signal peptide and prosequence. Where experimental data on PTM is not available, the use of a variety of bioinformatic tools available in the ExPASy Molecular Biology Server (www.expasy.org/proteomics) for predicting potential signal peptides, post-translational and proteolytic processing sites on poorly characterized proteins may be used to provide some guidance for peptide selection.

3. In general, it seemed that the antibodies were used for ANKRD1 characterization in tissue before experiments that are more basic were performed to validate the antibodies. I recommend at least one of three experiments for figure 4: 1) Repeat the two-site ELISA and western blotting in cell lines

where ANKRD1 is either overexpressed ectopically or endogenous ANKRD1 is removed by siRNA or CRISPR to show a differential signal, and/or 2) a competition experiment with free peptide to show loss of signal, and/or 3) overexpression of ANKRD1 with mutations in the key residues identified as critical for binding to show loss of signal. This would provide more confidence that they are actually measuring ANKRD1. This is particularly important where the recombinant protein is migrating at a different size on the western blot and their positive control - HPA038736 has not been validated for western blotting (<https://www.proteinatlas.org/ENSG00000148677-ANKRD1/antibody>). I would also recommend that their primary candidate for capture in the ELISA be assessed by immunoprecipitation followed by MS (IP-MS) to confirm that it actually does immunoprecipitate ANKRD1.

Response: We thank the reviewer for this constructive suggestion. We generated full length hANKRD1 cDNA and expressed the protein in *E. coli* and transfected H9c2 rat myoblast cells. We show that vector-alone transfected H9c2 cells expressed basal endogenous rANKRD1 and those transfected with the hANKRD1 cDNA showed increased ANKRD1 signals by ELISA and western blotting. ELISA data also showed good dilutional linearity and parallelism of lysates obtained from *E. coli* and H9c2 cells transfected with vector alone and vector carrying hANKRD1 (Figure 6). This shows reliable quantification following dilution of the different matrices and that antibody pair demonstrate similar binding responses towards endogenous rANKRD1 and vector-derived hANKRD1. Immunostaining of untransfected and transfected H9c2 cells also showed similar localization patterns as observed in rat heart tissue sections. In ELISA and immunostaining peptide competition experiments, successful blocking of rat and human ANKRD1 signals was achieved when matched peptides were used while signal retention was observed with unmatched peptides (Figures 6 and 7).

The capture antibody used in the ELISA AG5/mAb 4F2 was used for IP-MS from hANKRD1-expressing *E. coli*, neonatal rat cardiomyocytes and hANKRD1-transfected H9c2 cells. Mass spectrometry of tryptic digested pull-down material identified 6 hANKRD1 peptides from *E. coli* lysate. Unfortunately, ANKRD1 peptides were not identified in rat cell lysates, likely due to a combination of the relatively low level of ANKRD1 present compared with *E. coli* lysate, lower input rat lysate applied, and sensitivity of the MS instrument (Line 511-519).

4. Similar to point 3, for the IHC in figure 5, I would also recommend that the antibodies first be tested in a more controlled experiment such as in a cell line with ectopic ANKRD1 or depleted AKRD1 to show a differential signal before being tested in rat tissue. This is particularly important as AG1 showed nuclear staining in contrast to AG5 and the positive control HPA038736. I would also recommend a peptide competition experiment for the IHC to show the signal is specific to ANKRD1.

Response: We thank the reviewer for this constructive suggestion. As discussed above, western blot and ELISA data have shown increased signal from ectopic hANKRD1 over basal endogenous levels. In addition, the staining patterns obtained with H9c2 cells albeit having no sarcomeric structure showed very similar ANKRD1 cytoplasmic / nuclear localization with those observed in rat heart tissue. Hence, we feel that peptide competition for IHC is not necessary. Rather, a more useful specificity test is to show by western blotting that the mAbs do not bind to ANKRD2, a closely-related MARP family member that shares a high degree of sequence and functional similarity with ANKRD1. This confirmatory data has now been added in the main text (Lines 354-357, Supplementary Figure 8 and 9).

5. The authors clearly show the advantage of the DEXT microplates for this workflow. They should make it clear how readers could obtain these plates to replicate the workflow.

Response: DEXT microplates are the proprietary product of Dextech (S) Pte Ltd (www.dextechlab.com). They can be obtained for research testing and commercial applications by sending requests to peck@dextechlab.com. This information has been included in the methods section (Line 740).

Minor Comments:

1. Figure 2B – It is not entirely clear how the 14 peptides were selected. My assumption is that these are all those that meet the 0.35 score threshold (coloured yellow in A). Clarifying this would be easier for the reader to understand the progression from Figure 2A to 2B.

Response: Yes the 14 peptides were selected on the basis of the 0.35 threshold score. We have clarified this point in the legend to Figure 2.

2. Line 295 – It is not clear how the authors determined the ELISA assay detection limit as 0.073 ng/ml.

Response: The assay has a limit of detection (defined as the concentration derived from the mean optical density of 20 zero standard replicates plus 3 standard deviations) of 0.073 ng/ml. We have clarified this point in the text (Lines 277-278).

3. Figure 4C – the image is cropped on the right hand side. The HPA antibody is missing the 6 on the end.

Response: We thank the reviewer for spotting the mistake. We have corrected Figure 4C.

4. Line 610 – states “specific to one of the four test antigens” – should this be three (AG1, AG4, AG5)?

Response: In the primary screen, we tested the antibodies against three test antigens (Trx-AG1, Trx-AG4 and Trx-AG5). With fewer numbers to handle in the subsequent screening, we included an additional control antigen and tested the antibodies against four antigens (Trx alone, Trx-AG1, Trx-AG4 and Trx-AG5) as shown in Figure 3D. We have further clarified this in the text (Line 788) as well as the legend to Figure 3D.

5. In Supplementary Figure 4, the font size for the ELISA OD should be increased and I would recommend that they use a different color for the signal of the critical residues to better highlight them.

Response: We have made the necessary amendments of font size and colour for signal of the critical residues as suggested.

6. In Supplementary Table 2 – it would be helpful to clarify what the positive and negative control wells are.

Response: The negative control well is conducted exactly the same way as the corresponding positive well except that analyte is not added so that a sandwich capture-analyte-detector antibody

complex cannot be formed. This has been clarified in the legend to Supplementary Table 2.

Reviewer #3 (Remarks to the Author):

General comments:

The manuscript reads like two separate topics; (1) a description of a hybridoma-production pipeline (based on Figure 1), for generating, characterizing and validating mAbs, and (2) a validation study of several anti-hANKRD1 antibodies using said pipeline. Aside from the mash-up of these two topics, which is a serious issue (see below), the manuscript is well-written and would require little editing.

Response: We respectfully disagree with the reviewer's opinion that the two aspects outlined above represents separate topics mashed together. We strongly feel that in any solid work describing an efficient monoclonal antibody production method, full information on the hybridoma attrition rate to the final number of positive, high-yielding and stable mAb clones produced must be provided as an indication of efficient long term antibody production viability, along with comprehensive characterization and application-specific validation data of the positive mAbs to demonstrate their authenticity and utility spectrum. Only with these information combined will we be able to demonstrate the true success of the methodology described.

Because the manuscript is entitled "Efficient epitope-directed mAb production using a mixed Ag cocktail", I will focus my critique on this aspect of the two topics described above.

There is only a single example of the running of the pipeline. So the efficiencies, including the metrics for each step: percent success, successful applied to which applications, costs (and how costs are calculated) etc. are insufficient for anyone to adapt the method. Or even really to understand the demonstrated advantages toward using it. There is little innovation in the methods or pipeline described beyond possibly the use of trimer fusion peptides multiply injected into the animal.

Response: We respectfully disagree with the reviewer that the information provided is insufficient for anyone to adapt the method, to evaluate its efficiency or even to understand its advantages.

Firstly, the method know-how is fully described in the METHODS section.

Secondly, the metrics for method efficiencies is dependent on the objectives of each antibody project which in turn will define the acceptable performance requirement of the antibodies generated and thus the definition of success rate. It could be defined simply as the number of positive mAbs that are reactive to the peptide antigen, and we have provided clear information to show that 12 out of 12 (or 100%) of the positive mAbs do so. Alternatively, one may want to consider a more meaningful metric of the number of mAbs that are protein-reactive for which we provided clear information that 10 out of 12 (or 83.3%) of the mAbs recognized the native hANKRD1. Yet, another metric may be the number of positive mAbs that are protein-reactive and can be applied in both ELISA and western blot applications for which it was clearly stated that the success rate was 9 out of 12 (75%) positive mAbs. If efficiency is to be defined in terms of the total number of primary clones screened, we have indicated in the discussion section of the manuscript that one

protein-reactive mAb applicable for both Western blotting and ELISA is obtained from every 331 initial hybridoma clones screened. Hence, we chose to provide absolute numbers instead of percentages so that readers can evaluate success rates according to their own antibody project and antibody acceptance criteria.

Similar reasoning applies with working out cost efficiency of the method. We understand that the cost of reagents and labor varies between laboratories and countries. Hence, any researcher in the field would understand that immunization costs (animals, feed, housing and labor costs) could possibly be reduced by up to 3 folds by using a cocktail of three antigens as immunogen in one hybridoma production cycle compared with multiple single-antigen immunization schemes. The use of novel DEXT microplates that affords assay volume miniaturization to 15 μ l instead of the typical 100 μ l volumes with standard 96-well microplates would translate to reagent cost savings of up to 6 folds. Furthermore, the use of DEXT microplates for hybridoma clone screening allows direct visual identification of positive clones upon addition of enzyme substrate without the need for stopping the reaction (savings in reagent cost and labor cost) as well as reading the absorbance (savings in time which would translate to savings in labor cost). Clearly, there are many areas in the whole antibody production process where cost savings are difficult to quantify exactly. However, sufficient information has been provided in the manuscript for each reader to estimate their own cost savings for the antibody production steps outlined above. In addition, to our knowledge there are no published reports that provide detailed cost calculations for the antibody production method described in their studies.

We address the reviewer's comment that insufficient information was provided for readers to understand the demonstrated advantages for using our described method. We point to Figure 1 of the manuscript which provides a clear overview of the advantages/purpose of each step of the antibody production/characterization and validation workflow.

We agree with the reviewer that the method was demonstrated with one model protein and it would be ideal to provide further evidence of method generality with a second test protein. However, this is not a trivial exercise requiring substantial resources/funding and at least 1.5 years to complete. Nevertheless, we have addressed this issue in the manuscript (Lines 676-691) by elaborating on our previous success whereby the same trimer peptide fusion strategy led to breakthrough generation of mAbs that are specific to single amino acid p53 mutation hotspots - R175H, R248Q and R273H (Cell Reports, 2018, 22, 299-312). All three mutant specific mAbs detected their corresponding mutant proteins without cross reacting to other mutant p53 proteins or to the wild-type p53 protein. The fact that the R248Q mAb was unable to detect the R248W mutant protein further highlighted the specificity of the mAb generated. The primary differences in this earlier p53 work compared with the present manuscript are:

- 1) Each mouse was immunized with a single mutant peptide antigen
- 2) There was no scope for epitope prediction in this early work since the peptide antigen used was dictated by the location of the mutation hotspot on the p53 sequence. As such, all the recombinant p53 mutant trimer fusion proteins produced in *E. coli* were primarily expressed in insoluble inclusion bodies which necessitated extra refolding work to obtain soluble proteins in physiological buffer suitable for immunization. Protein refolding is a time consuming and tedious task. It is especially challenging to be able to achieve refolded

proteins at high concentrations (>0.5 mg/ml) to minimize animal discomfort during immunization.

Herein lies the benefits of our current work where we show it is possible to combine multiple antigens in a single cocktail for immunization. In addition, where there is more flexibility in antigen peptide selection, the use of epitope prediction guides the selection of peptides not only with higher likelihood of being an effective epitope but also with physicochemical properties associated with soluble bacterial expression as exemplified in our work where all selected antigens were highly expressed primarily as soluble proteins in *E. coli*. This has allowed all three antigens to be easily purified by one-step IMAC to concentrations of at least 1 mg/ml each. This additional information has now been added to the discussion in the main text.

We are confident that our trimer peptide fusion strategy represents a highly effective way of antigen presentation that consistently leads to successful generation of desired mAbs of high specificity and affinity. In fact, since our first report of mAbs against R175H, R248Q and R273H mutations, we have progressed successfully to generate specific mAbs to other p53 mutations; Y220C, G245S, R282W and R248W (unpublished data in collaboration with DP Lane's p53 research group). Hence, our additive improvements to this antigen presentation strategy by introducing epitope prediction, careful peptide selection against background knowledge of protein structure/PTM/protein interaction sites where available, use of a cocktail immunogen and hybridoma screening with novel DEXT microplates described in this current manuscript are geared effectively towards obtaining 'protein-reactive' peptide antibodies and streamlining downstream mAb selection, characterization and application-specific validation.

But without a lot more data describing the benefits to speed, quality or cost. The audience for this information would be other users or developers of an Ab pipeline. These would be sophisticated researchers. And the data they would need to evaluate this pipeline is missing.

Response: We have now provided additional information on the manpower and time taken for epitope mapping by alanine scanning (Lines 655-656).

In the discussion section, we have also provided additional comparative information on the functional quality of the mAbs we produced based on binding affinity data (Lines 533-541). We extracted the available curated binding affinity data of 87 mouse monoclonal antibodies from the SabDab structural antibody database (Dunbar et al. 2014, *Nucleic Acids Research*, 42, D1140-D1146) and used the published KD values to classify their binding strength as low (micromolar range; 10^{-5} to 10^{-6} M), moderate (nanomolar range; 10^{-7} to 10^{-9} M), high (picomolar range; 10^{-10} to 10^{-11} M) and very high (10^{-12} M). This data provides an idea of the typical quality of antibodies since the affinity requirement of functional antibodies lies in the nanomolar range. As shown in the pie-chart, 82% of the antibodies have moderate binding affinity.

In contrast, 10 out of 12 peptide antibodies (83%) produced in our work were found to have high binding affinity to the native protein in the picomolar (10⁻¹⁰ to 10⁻¹¹ M) range while two mAbs showed binding only to the peptide and not the native protein. Additionally, the protein-reactive binders can be used in immunoassay, western blotting and cell staining applications. Hence, convincing evidence has been presented in our work demonstrating the ability of the designed peptide antigens to produce a high percentage of quality mAbs that are fit-for-purpose.

The work on the isolation and validation of the hANKRD1 is fine. But again, that's not the focus of the paper.

Response: We feel that a full description of the isolation and validation of the Trx-hANKRD1 peptide antigens is important since demonstration of facile production of high quality and authentic antigens is fundamental to any antibody production method.

Reviewers' comments:

Reviewer #1 (Remarks to the Author):

In this paper, mixed antigens were used to produce monoclonal antibody. The author did lots of work to conduct this research. Importantly, this method facilitates development of multiple distinct antibodies that can provide further confidence in antibody specificity and enable assays such as two-site ELISAS. However, relevant references and data of some experiment need to be supplied. This paper will need a major revision.

1. The manuscript is entitled "Efficient epitope-directed mAb production using a mixed antigen cocktail", the research focus on two main topics: antibody production via the novel method and validation via different experiments, the author should stress the highlight of the research.
2. The "MIAN" part should be rewritten, and the research purpose of this study described here is not very clear.
3. The authors show the advantage of the DEXT microplates, but did not make it clear to reader what the detailed process of this part, and also in the "MAIN" part, the author need supply some introduction of the DEXT microplates application.
4. This is a novel antibody production technology, there should be some comparative studies with traditional hybridoma technology in the research, and emphasizing the innovation of this research by supplying some related data.
5. In the discussion, the results show that very short epitope-predicted peptides spanning 13-24 residues can be used successfully to produce mAbs, the author should find some other research as a comparative discussion.
6. The Figure 8 is the mechanistic scheme of ANKRD1 cellular localization. Is it important for this new antibody production research?
7. The two-site sandwich ELISA is also an important application of the antibody. The author should discuss and introduce this part in the manuscript.
8. The quality of figures needs to be improved.
9. Line 255: the "mAB" in the header of table 1 should be revised as "mAb", and also the The font of the table text needs to be consistent with the manuscript text.
10. The author should add a ruler in the figure 5.
11. The format of the manuscript, especially the references needs to be careful revised.

Reviewer #2 (Remarks to the Author):

I would like to thank the authors for providing the additional experiments that I suggested. I am convinced that their antibodies are recognizing purified human ANKRD1. The challenge for many antibodies and where they often fail is being able to specifically and sensitively recognize endogenous proteins. Unfortunately, from the evidence provided in this revised manuscript, I remain unconvinced that the antibodies are recognizing ANKRD1 from lysate or tissue. As they are basing their entire validation of their pipeline on antibodies to ANKRD1 (and prior publication with p53), I do not feel that they have provided enough evidence that their pipeline is effective to warrant publication.

Major Points

1. The fact that their IP-MS did not detect ANKRD1 in rat lysate makes me concerned that their capture antibody in ELISA may not actually be identifying rANKRD1. Starting material of >1 mg of total protein in a biotin/strep pulldown with analysis on a 5600 should be sensitive enough to pick up what is essentially the 'bait'. The failure to detect rANKRD1 in this experiment indicates that the antibody may not be recognizing the endogenous rANKRD1 protein. For the IP-MS of hANKRD1 from E. coli and starting with 14 mg of starting material, I would have expected more complete coverage of the 'bait' rather than just 6 peptides also suggesting that the antibody may not recognize hANKRD1 that effectively.
2. In figure 6, the western blot is convincing that the 3 antibodies are detecting purified protein (lanes

1 and 4). For 30D12 – I can also see a band for ectopic expressed hANKRD1 in lane 3 but for the other two, there may be a slight increase over the background bands. Probing with myc would give an idea of where the band is and what is the differential in signal that we would expect to see between lanes 2 and 3.

3. In figure 4, they indicate that the HPA antibody does not work in western blotting in the Human Protein Atlas or for endogenous rANKRD1 so I do not see a positive control defining where rANKRD1 actually runs on a western blot. Even with the HPA antibody, I still see the background bands with different intensities in some of the lanes. Each antibody (all run on different gels and sometimes in different loading order making comparison challenging) being tested seem to give a different ratio of signal in S compared to N. This data does not convince me that endogenous rANKRD1 is being identified.

4. For the IF in figure 7, it is reassuring that the signal can be competed away with the matched peptide but I would have expected to see an increase in signal when adding hANKRD1 which I do not see and with the unconvincing IP-MS and western blot results, I wonder if the different localizations of ANKRD1 with the different antibodies may be non-specific binding.

Minor Point

5. The authors should consider changing the flow of their manuscript as it makes more sense to add the validation of their antibodies (Figure 6) before they probe for endogenous rANKRD1 in figure 4. I would also recommend moving figure 4a into figure 6 as it is showing linearity for recombinant ANKRD1.

Response to Reviewers

We are pleased to resubmit our revised manuscript entitled “Efficient Epitope-directed Monoclonal Antibody Production Using A Mixed Antigen Cocktail Facilitates Antibody Characterization and Validation”. We have carefully considered each concern and where applicable have made the necessary changes to the manuscript (highlighted in blue for easy tracking). Detailed point-by-point responses are addressed below:

Reviewer #1 (Remarks to the Author):

In this paper, mixed antigens were used to produce monoclonal antibody. The author did lots of work to conduct this research. Importantly, this method facilitates development of multiple distinct antibodies that can provide further confidence in antibody specificity and enable assays such as two-site ELISAS. However, relevant references and data of some experiment need to be supplied. This paper will need a major revision.

1. The manuscript is entitled “Efficient epitope - directed mAb production using a mixed antigen cocktail”, the research focus on two main topics: antibody production via the novel method and validation via different experiments, the author should stress the highlight of the research.

Response: We have amended the title of the manuscript to reflect both the novel antibody production method and its facilitation of antibody characterization and validation.

2. The “MIAN” part should be rewritten, and the research purpose of this study described here is not very clear.

Response: As required by the journal format, the heading “MAIN” has now been replaced by “INTRODUCTION”.

The “INTRODUCTION” section was logically developed by first outlining the antibody challenge in the opening paragraph and ending with a clear statement of the objective of the study – to develop an efficient peptide-mediated method for producing fit-for-purpose antibodies (Lines 56-58) and highlighting that the workflow reduces the experimental burden for hybridoma screening, antibody characterization and validation (Lines 58-60).

We have now strengthened this section by describing the advantages of using peptides as immunogens (Lines 65-71) with additional references, and clarified how the antigen design strategy facilitates immunogen production and purification, epitope mapping and two-site ELISA development (Lines 80-96). We have also added a final paragraph in the “INTRODUCTION” section to summarize the findings as required by the journal (Lines 109-116).

3. The authors show the advantage of the DEXT microplates, but did not make it clear to reader what the detailed process of this part, and also in the “MAIN” part, the author need supply some introduction of the DEXT microplates application.

Response: A brief description of the DEXT microplate was provided in the third paragraph of the “INTRODUCTION” section with three references that direct readers to more information on concept design and working principles of the DEXT microplate (Lines 84-88). The DEXT microplate assembly is

clearly shown in Figure 3a and further details on its usage for hybridoma screening are also provided in the RESULTS (Lines 136-142) and METHODS (Lines 812-863) section of the manuscript as well as Supplemental Materials (pages 4 and 5).

For improved clarity, we have replaced the wording “A bespoke receptacle” to **DEXT microplates** in the “INTRODUCTION” section (Line 84) and highlighted its advantages over conventional microplates (Lines 86-87).

4. This is a novel antibody production technology, there should be some comparative studies with traditional hybridoma technology in the research, and emphasizing the innovation of this research by supplying some related data.

Response: Comparative data on antibody generation efficiency against a large consortia effort using hybridoma-based methods or phage display approaches was provided in the Discussion section (Lines 556-564). We also provided typical affinities of curated antibodies deposited within the SabDAB structural antibodies database and found that 82% of these were of low to moderate affinity. In contrast, we emphasized that with our method, 83% of the mAbs displayed binding affinities in the high picomolar range (Lines 565-572).

5. In the discussion, the results show that very short epitope-predicted peptides spanning 13-24 residues can be used successfully to produce mAbs, the author should find some other research as a comparative discussion.

Response: We find it difficult to make direct comparative analyses of the efficiency of our antibody production method with other reports of generating anti-peptide antibodies. The metrics for evaluating method efficiencies varies with each antibody project and most authors do not provide comparable metrics to allow meaningful comparison. For example, Fieser et al. (1987, PNAS, 84, 8568-8572) reported 80% of their hybridomas (34/43) generated anti-peptide mAbs that reacted to both the original protein and the peptide antigen during initial screening. However, there is no data as to how many of these clones were stable and high yielding after subsequent cycles of sub-culture which is an essential metric for evaluating the true success rate of the antibody production method.

Nevertheless, we have now provided some comparative data of the relative success of our method with other reports in producing protein-reactive antibodies using peptides as the immunogen (Lines 571-574).

6. The Figure 8 is the mechanistic scheme of ANKRD1 cellular localization. Is it important for this new antibody production research?

Response: As highlighted in the “INTRODUCTION” section (Lines 105-108), the ability to produce well- characterized antibodies that bind to different functional domains of a target protein will facilitate mechanistic understanding of its biology. The addition of Figure 8 illustrates the point by deriving a mechanistic scheme for ANKRD1 subcellular localization. To improve the flow of the manuscript, we have moved this discussion towards the end of the “DISCUSSION” section.

7. The two-site sandwich ELISA is also an important application of the antibody. The author should discuss and introduce this part in the manuscript.

Response: We have now highlighted in the "INTRODUCTION" section that the method overcomes challenges in two-site immunoassay development by generating antibodies to multiple non-overlapping sites on the protein (Lines 93-96). In the Discussion section, we have also mentioned the value of our method to produce mAbs targeting spatially distant sites on the target protein to obtain matched antibody pairs for two-site ELISA (Lines 598-600) and the implications of unexpected self-pairing of two mAbs that portend opportunities to measure dimeric forms of circulating ANKRD1, a tantalizing prospect that could open new vistas in diagnostic biomarker development (Line 736-743). With word limitations for this journal, we feel there is a need to be succinct and sufficient mention of the key points relevant to ELISA application has already been provided.

8. The quality of figures needs to be improved.

Response: Some loss in resolution of the figures occurred during conversion of the original image files to pdf format. This problem will be resolved when image files are submitted in pptx format if this manuscript is accepted for publication.

We have also made adjustments and improvements to all the figures for enhanced clarity.

9. Line 255: the "mAB" in the header of table 1 should be revised as "mAb", and also the The font of the table text needs to be consistent with the manuscript text.

Response: The typo and font of the table has been corrected as suggested.

10. The author should add a ruler in the figure 5.

Response: We have added a ruler to Figures 6 and 7 (change of Figure numbering due to adjustments to flow of manuscript).

11. The format of the manuscript, especially the references needs to be careful revised.

Response: We have adjusted the flow of the manuscript and revised the references in accordance with the journal's format.

Reviewer #2 (Remarks to the Author):

I would like to thank the authors for providing the additional experiments that I suggested. I am convinced that their antibodies are recognizing purified human ANKRD1. The challenge for many antibodies and where they often fail is being able to specifically and sensitively recognize endogenous proteins. Unfortunately, from the evidence provided in this revised manuscript, I remain unconvinced that the antibodies are recognizing ANKRD1 from lysate or tissue. As they are basing their entire validation of their pipeline on antibodies to ANKRD1 (and prior publication with p53), I do not feel that they have provided enough evidence that their pipeline is effective to warrant publication.

Response: We thank the reviewer for acknowledging that our mAbs recognize hANKRD1. While we agree that more work can be done to provide additional supporting data, we feel that the body of evidence from antibody characterization and validation by ELISA (supported by parallelism and peptide competition analysis), Western blotting, immunocytochemistry (supported by peptide competition analysis) as well as immunoprecipitation provides more than minimal documentation to demonstrate an acceptable level of confidence in the specificity of our mAbs towards endogenous ANKRD1. This is a paper that describes an antibody production method starting from antigen design and it is not reasonable to expect the level of exhaustive validation needed for more definitive and conclusive evidence of mAb specificity for some applications. This is well illustrated by the work of Anderson et al. ("Insufficient antibody validation challenges oestrogen receptor beta research", Nature Communications, 8, 15840, 2017) whereby detailed validation of pre-existing mAbs from various commercial sources just for IHC application alone takes up an entire paper in a high impact factor journal.

We are confident that the pipeline of ANKRD1 mAbs and prior work on p53 mAbs provides sufficient evidence of the effectiveness of our peptide-mediated antibody production method. Taken together, the method has been demonstrated to be robust and works consistently: 3 sites for ANKRD1, 3 sites for p53 mutation hotspots plus a further 4 sites yet to be published, all in all representing 10 out of 10 different peptides or 100% success in generating mAbs that are protein-reactive and showing reasonable specificity and utility in a number of applications. It is also evident that our antigen-presentation strategy is immune to whether or not there is broad flexibility in peptide selection. This is well illustrated in the p53 mutant mAbs where peptide selection is largely dictated by the position of the amino acid mutation and yet mAbs specific to single amino acid mutations can be generated consistently. This portends utility of our antibody production method for a broad repertoire of targets.

Major Points

1. The fact that their IP-MS did not detect ANKRD1 in rat lysate makes me concerned that their capture antibody in ELISA may not actually be identifying rANKRD1. Starting material of >1 mg of total protein in a biotin/strep pulldown with analysis on a 5600 should be sensitive enough to pick up what is essentially the 'bait'. The failure to detect rANKRD1 in this experiment indicates that the antibody may not be recognizing the endogenous rANKRD1 protein. For the IP-MS of hANKRD1 from E. coli and starting with 14 mg of starting material, I would have expected more complete coverage

of the 'bait' rather than just 6 peptides also suggesting that the antibody may not recognize hANKRD1 that effectively.

Response: We agree with the reviewer that successful IP-MS detection of ANKRD1 in rat lysate would have added a greater level of confidence on antibody specificity towards endogenous rANKRD1. ELISA data based on the 4F2/30D12 antibody pair showed that the concentration of ANKRD1 in pCMV-Myc/hANKRD1 lysate (2mg/ml total protein) was 10.3 ng/ml. This meant that 1 mg of total protein was expected to contain only about 5 ng of ANKRD1. Due to multiple steps of pull-down and sample processing before LC/MS analysis, which inevitably resulted in sample loss, it is not surprising that the final input of ANKRD1 digested peptides could fall below the detection limit by MS.

Nevertheless, we were able to identify 6 hANKRD1 signature peptides by IP-MS which corresponds to 20% sequence coverage. We also understand that the detectability of a peptide by any MS instrument varies considerably and are strongly influenced by the chemical properties of the particular sequence, hence affecting peptide ionization (Vogel and Marcotte, *Nature Protocols*, 3(9), 1444, 2008; Steen and Pandey, *Trends in Biotechnology*, 20,361-364, 2002). It is thus speculative to attribute the lack of complete sequence coverage in IP-MS solely to poor recognition of the antibody to its target.

2. In figure 6, the western blot is convincing that the 3 antibodies are detecting purified protein (lanes 1 and 4). For 30D12 – I can also see a band for ectopic expressed hANKRD1 in lane 3 but for the other two, there may be a slight increase over the background bands. Probing with myc would give an idea of where the band is and what is the differential in signal that we would expect to see between lanes 2 and 3.

Response: We have provided the western blot probed with an anti-Myc antibody (Figure 5b) to show differential detection at the expected 40-kDa band position of ectopic hANKRD1.

3. In figure 4, they indicate that the HPA antibody does not work in western blotting in the Human Protein Atlas or for endogenous rANKRD1 so I do not see a positive control defining where rANKRD1 actually runs on a western blot. Even with the HPA antibody, I still see the background bands with different intensities in some of the lanes. Each antibody (all run on different gels and sometimes in different loading order making comparison challenging) being tested seem to give a different ratio of signal in S compared to N. This data does not convince me that endogenous rANKRD1 is being identified.

Response: The literature reports that rANKRD1 migrates at the ~40-kDa position in rat heart muscle (ref 32) and rat neonatal cardiomyocytes (ref 33, 34) and this is mentioned in the manuscript (Lines 395-399). In close agreement, the immunodetected rANKRD1 bands in rat heart tissue lysate and neonatal cardiomyocytes (now in Figure 5a) were 39 ± 0.7 kDa and 41 ± 0.3 kDa respectively as determined by densitometric analysis. Hence, we have corroborative evidence for immunodetection of endogenous rANKRD1 in these samples. Also, the presence of a 40-kDa band in pCMV-Myc transfected H9c2 lysate observed for AG4/mAb 5C1 and AG5/mAb 4F2 (Figure 5c) would indicate that endogenous rANKRD1 runs at the same position as ectopic hANKRD1 in pCMV-Myc/hANKRD1-transfected H9c2 lysate.

It is noteworthy that unlike previous reports, we ran rat heart tissue and rat neonatal cardiomyocyte lysates side-by-side on the same blot which revealed a slight molecular weight shift between the two and more work could be done to elucidate the nature of protein modification responsible for this mobility shift. Our WB data shows that the relative band size and ratio of band intensities obtained with AG1 (except the IgM 8A3 mAb which has a mismatch in the critical binding residue) and AG5 mAbs are generally consistent. The inconsistent banding profile primarily resides with AG4 mAbs and this does not necessarily indicate non-specificity of the antibodies but rather may be a reflection of some structural or biological impediment to antibody interaction with the internal domain of ANKRD1. We recognize that more work to clarify this is warranted in future work.

For the HPA antibody, densitometric analysis of the two bands observed with rat heart tissue lysate indicates an offset position at ~42.5 and 51.8 kDa position. Since we were not able to observe the expected immunodetection of recombinant hANKRD1 with this antibody, we are not able to make any statement regarding the immunodetected bands in rat heart tissue lysate by this antibody.

We apologize for the inconsistent loading sequence for some of the mAb panels.

4. For the IF in figure 7, it is reassuring that the signal can be competed away with the matched peptide but I would have expected to see an increase in signal when adding hANKRD1 which I do not see and with the unconvincing IP-MS and western blot results, I wonder if the different localizations of ANKRD1 with the different antibodies may be non-specific binding.

Response: We appreciate the scientific rigor expressed by the reviewer. Ideally, the IF experiment should be performed in a cell line that does not normally express endogenous ANKRD1 but unfortunately this was not the case for H9c2 cells. As shown by Western blotting, endogenous rANKRD1 is present in H9c2 cells. The modest increase in the 40-kDa band detected by the 5C1 and 4F2 mAbs in pCMV-Myc/ANKRD1 lysate relative to pCMV-Myc indicates low ectopic expression of hANKRD1. Hence, it would be tenuous to expect visually discernible increase in fluorescence signal over a strong background of endogenous rANKRD1.

We would also like to point out that it is commonly accepted that competition or adsorption controls are the minimum acceptable documentation in support of antibody specificity in IHC (Saper and Sawchenko, 2003, *J. Comp. Neurol.* 465(2), 161-163), on condition that the antigen used for immunization does not share common sequences with closely-related or other known proteins. We have presented sufficient primary data that our hANKRD1 mAbs are capable of recognizing endogenous rANKRD1 on the following basis:

1. The subcellular location of rANKRD1 in the nucleus and sarcomere in rat cardiomyocytes (Figure 7) is supported in the literature. We show that AG1/mAb 30D12 stains both nucleus and sarcomere and this agrees with previous reports (ref 32 and 33 and discussed in Lines 651-654) where the same localization pattern is obtained with rabbit polyclonal antibodies raised against the N-terminal fragment of ANKRD1. In addition, the cytoplasmic localization of rANKRD1 probed with AG5/mAb 4F2 is in full agreement with the IHC-validated rabbit polyclonal antibody HPA038736 which is raised against the C-terminal domain of hANKRD1 (Lines 655-656). Thus we have strong evidence showing corroborative staining patterns by other independent antibodies raised against similar regions as the 30D12 and 4F2 mAbs.

2. Orthogonal evidence from sustained upregulation of ANKRD1 mRNA expression in response to cardiac insults reported in the literature (reviewed in Ling et al. 2017, International Journal of Molecular Science, 18, 1362) and this is corroborated by increased immunostaining by AG1/mAb30D12 and AG5/mAb4F2 in the border region of infarcted rat tissue relative to the remote region shown in Figure 7.
3. As pointed out earlier, our mAbs detected the 40-kDa rANKRD1 band in Western blots which is in agreement with that reported in the literature. In addition, we also show that mAbs directed to non-overlapping regions of ANKRD1 (AG1: 30D12; AG5: 4F2, 6A4 and 11D11) show similar western blot banding patterns in rat cardiac tissue and rat neonatal cardiomyocytes.
4. We have presented primary data from peptide competition experiments in ELISA and IF to provide an acceptable level of confidence in the specificity of the mAbs for their cognate ANKRD1 peptide sequences. The well characterized antibody pair of 30D12 and 4F2 also works in a two-site assay measuring ANKRD1 in rat H9c2 and *E.coli* lysate which shows good parallelism when compared against hANKRD1. This provides sound basis for confidence of the selectivity of the antibodies for rANKRD1 and hANKRD1. Parallelism also demonstrates similarity of the immuno-affinity characteristics between the rANKRD1 and hANKRD1.
5. Our mAbs do not cross-react with the closely-related family member, hANKRD2.
6. Blast information does not give any indication that the selected peptide antigens have sequence similarities with other protein targets.

The results from Western blot and ICC although not perfect gives a reasonable degree of assurance that the mAbs recognize endogenous ANKRD1. We do agree with the reviewer that further work could be done to provide more definitive data and have thus included a qualifying statement in the manuscript (Line 762-765).

Minor Point

5. The authors should consider changing the flow of their manuscript as it makes more sense to add the validation of their antibodies (Figure 6) before they probe for endogenous rANKRD1 in figure 4. I would also recommend moving figure 4a into figure 6 as it is showing linearity for recombinant ANKRD1.

Response: We thank the reviewer for this constructive suggestion. We have made the adjustments to the relevant figures and flow of the manuscript as suggested.

REVIEWERS' COMMENTS:

Reviewer #2 (Remarks to the Author):

I'd like to thank the authors for carefully reviewing and responding to my concerns and for clarifying in the ELISA section the low amount of ANKRD1 present in cells. This helps with expectations about what to expect for western and MS analysis. The authors have chosen a challenging protein to showcase their antibody production pipeline which helps highlight the strength of their assay but also makes it harder to clearly show their validation.

I focused this review on the changes made to address my concerns and my only remaining concern is related to the immunoblotting in Figure 5b. I have made the following suggestions to help:

1) Looking at figure 5b – with low expression of hANKRD1 there are background bands that are appearing in the anti-myc blot but there is a distinct band when ANKRD1 is expressed in lane 3 (note the legend is scrambled lines 460/461 – so the anti-myc blot is not well described). I agree with the authors that an equivalent band is seen with 30D12 in lane 3. With 5C1 and 4F2 – as the authors mention there may be an increase in signal in lane 3 with ectopic expression but to conclude anything – this would have to be repeated multiple times and have the band quantified. Given the multiple bands seen of equivalent intensity with 5C1 and 4F2 – I would not conclude that one of these bands is endogenous ANKRD1 because it migrates at the expected position.

2) My suggestion would be to simplify figure 5 – keep the first two panels of b) (anti-myc and 30D12) and then keep the panel of 30D12 from A) – final figure would be 3 images. Put the other data in a supplementary. In this supplementary put the data under two columns – “signal consistent with ANKRD1 detection” and “no signal consistent with ANKRD1 detection” (or some title to that effect) and say that these antibodies would need further validation.

With these changes, the paper would be acceptable to me for publication.

Response to Reviewers

We are grateful to all reviewers for their insightful and constructive suggestions to enhance the rigor and improve the quality of our manuscript. We are pleased to resubmit our final revised manuscript entitled “Epitope-directed Monoclonal Antibody Production Using A Mixed Antigen Cocktail Facilitates Antibody Characterization and Validation”. We have carefully considered the final concerns and have made the necessary changes to the manuscript.

Reviewer #2 (Remarks to the Author):

I'd like to thank the authors for carefully reviewing and responding to my concerns and for clarifying in the ELISA section the low amount of ANKRD1 present in cells. This helps with expectations about what to expect for western and MS analysis. The authors have chosen a challenging protein to showcase their antibody production pipeline which helps highlight the strength of their assay but also makes it harder to clearly show their validation.

Response: We thank the reviewer for the positive and encouraging comments.

I focused this review on the changes made to address my concerns and my only remaining concern is related to the immunoblotting in Figure 5b. I have made the following suggestions to help:

1) Looking at figure 5b – with low expression of hANKRD1 there are background bands that are appearing in the anti-myc blot but there is a distinct band when ANKRD1 is expressed in lane 3 (note the legend is scrambled lines 460/461 – so the anti-myc blot is not well described). I agree with the authors that an equivalent band is seen with 30D12 in lane 3. With 5C1 and 4F2 – as the authors mention there may be an increase in signal in lane 3 with ectopic expression but to conclude anything – this would have to be repeated multiple times and have the band quantified. Given the multiple bands seen of equivalent intensity with 5C1 and 4F2 – I would not conclude that one of these bands is endogenous ANKRD1 because it migrates at the expected position.

*Response: For Figure 5b, we agree with the reviewer that the bands in lane 2 probed with 5C1 and 4F2 cannot be definitively concluded to represent endogenous ANKRD1 simply on the basis that they migrate at the expected molecular weight position. However, it should be acknowledged that when western blotting is used as a first step in antibody validation, it is widely accepted that an indication of antibody specificity is the observation of a single band at the expected molecular weight position; albeit not definitive but suggestive of reasonable likelihood. In addition, the presence of other immunodetected bands at higher or lower molecular weight positions does not necessarily indicate non-specificity but rather could represent post-translationally modified targets, splice variants or proteolytic breakdown entities, thus, suggesting that other confirmatory experiments be performed (Bordeaux et al. Antibody Validation. *Biotechniques* 2010: 48(3); 197-209). Given that both ELISA and IHC data indicate the presence of endogenous rANKRD1 in H9c2 cells, it is reasonable to expect that the 40-kDa band in lane 2 of the 5C1 and 4F2 probed blots in all likelihood represent endogenous rANKRD1. Nevertheless, we agree with the reviewer to take a cautious approach in interpreting the data and have made the following amendments in the “Results” and “Discussion” section as follows:*

“Surprisingly, with AG4/mAb 5C1 and AG5/mAb 4F2, an immunodetected band that coincides with the 40-kDa position of full-length endogenous rANKRD1 was detected in pCMV-Myc transfected

lysate while intensity of the equivalent band in pCMV-Myc /hANKRD1 lysate was slightly higher, indicative of ectopic hANKRD1 expression.” (Pg 11, para 1, lines 3-7).

“Intriguingly, Western blot data showed that AG1/mAb 30D12 failed to detect endogenous full-length rANKRD1 in H9c2 cell lysate although its presence **may be indicated** by the positive 40-kDa bands obtained with AG4/mAb 5C1 and AG5/mAb 4F2.” (Pg 17 Lines 24 – Pg 18 Line 1)

2) My suggestion would be to simplify figure 5 – keep the first two panels of b) (anti-myc and 30D12) and then keep the panel of 30D12 from A) – final figure would be 3 images. Put the other data in a supplementary. In this supplementary put the data under two columns – “signal consistent with ANKRD1 detection” and “no signal consistent with ANKRD1 detection” (or some title to that effect) and say that these antibodies would need further validation.

Response: We thank the reviewer for this constructive suggestion to simplify Figure 5. We feel that all through the manuscript we have been presenting data for at least one representative antibody from each antigen group. To show data only for the 30D12 antibody in Figure 5 as suggested runs counter to consistency in the manuscript. Hence, we have modified Figure 5 to only show data for 30D12, 5C1 and 4F2 and moved all other WB data to the Supplementary Information (Supplementary Figure 7 and 8). In Supplementary Figure 7, we have also classified the panel of mAbs under 3 categories:

1. Antibodies detecting bands consistent with recombinant hANKRD1 and endogenous rANKRD1.
2. Antibodies detecting bands consistent with recombinant hANKRD1 but not endogenous rANKRD1.
3. Antibodies that do not detect bands consistent with recombinant hANKRD1 or endogenous rANKRD1.

With these changes, the paper would be acceptable to me for publication.

Response: We trust that we have now adequately addressed all the concerns of the reviewer.